# Two components of body-image disturbance are differentially associated with distinct eating disorder characteristics in healthy young women

Yumi Hamamoto[1,2]*, Shinsuke Suzuki[2,3,4], Motoaki Sugiura[2,5]

1 School of Medicine, Tohoku University, Sendai-shi, Miyagi, Japan, 2 Institute of Development, Aging and Cancer, Tohoku University, Sendai-shi, Miyagi, Japan, 3 Frontier Research Institute for Interdisciplinary Sciences, Tohoku University, Sendai-shi, Miyagi, Japan, 4 Brain, Minds and Markets Laboratory, Department of Finance, The University of Melbourne, Carlton Victoria, Australia, 5 International Research Institute of Disaster Science, Tohoku University, Sendai-shi, Miyagi, Japan

* yumi.hamamoto.q2@dc.tohoku.ac.jp, yumi.hamamoto.q2@gmail.com

## Abstract

Body-image disturbance comprises two components. The first is perceptual in nature, and is measured by a discrepancy between one's actual body and perceived self-image ("perceived–actual discrepancy"). The other component is affective, and is measured by a discrepancy between one's perceived self-image and ideal body image ("perceived–ideal discrepancy"). The present study evaluated the relationships between body-image disturbance and characteristics of eating disorders such as symptoms and related personality traits. In a psychophysiological experiment, female university students (mean ± SD age = 21.0 ± 1.38 years) were presented with silhouette images of their own bodies that were distorted in terms of width. The participants were asked whether each silhouette image was more overweight than their actual or ideal body images. Eating-disorder characteristics were assessed using six factors from the Japanese version of the Eating Disorder Inventory 2 (EDI2). We found that perceived–actual discrepancies correlated with negative self-evaluation (i.e., factor 3 of the EDI2), whereas perceived–ideal discrepancies correlated with dissatisfaction with one's own body (i.e., factor 2 of EDI2). These results imply that distinct psychological mechanisms underlie the two components of body-image disturbance.

## 1 Introduction

Body-image disturbance refers to a negative evaluation of, and attitude toward, one's own body, which is widely considered a core symptom of eating disorders [1, 2]. Healthy individuals with greater body-image disturbance show higher levels of eating-disorder-related characteristics, such as interoceptive awareness, ineffectiveness, bulimia, body dissatisfaction, and drive for thinness [3, 4]

**Data Availability Statement:** All relevant data are within the manuscript and its Supporting Information files.

**Funding:** This study was supported by KAKENHI 16H01873 (MS) and 17H05933 (SS) from Japan Society for the Promotion of Science (https://www.jsps.go.jp/english/index.html). YH received the financial support from the Division for Interdisciplinary Advanced Research and Education Selective Examination, Tohoku University (http://www.iiare.tohoku.ac.jp/en/). The funders had no role in study design, data collection and analysis, decision to publish, or preparation of the manuscript.

**Competing interests:** The authors have declared that no competing interests exist.

Accumulating evidence implies that body-image disturbance consists of perceptual and affective components [5–7]. Several studies have also suggested that body-image disturbance has a cognitive component, which is sometimes considered to overlap with the affective component and is referred to as the "cognitive-affective component" [8–10]. Moreover, a neuroimaging study of body-image disturbance suggested that the cognitive and affective components are not independent. That study categorized previous neuroimaging studies based on the body-image disturbance component that was evaluated, and reported that studies investigating the affective or cognitive component reported brain activation in similar regions [6].

Perceptual and affective components are conceptually different. The perceptual component refers to a disturbance in the perception of one's own body; patients and healthy people exhibiting this component are unable to appraise their bodies accurately [11–13]. For example, people with eating disorders and healthy individuals with more eating-disorder-related characteristics tend to overestimate their body size (i.e., there is a discrepancy between the perceived and actual self-body sizes) [11, 12, 14]. The affective component relates to a disturbance in attitudes and feelings toward one's own body. Patients and healthy people who exhibit the component are extremely dissatisfied with their body, i.e., their ideal and perceived body images differ considerably [11, 12, 14]. Functional magnetic resonance imaging (fMRI) studies of people with eating disorders have indicated that two components of body-image disturbance relate to different phases of body-image processing: the perceptual component relates to visual processing (input), while the affective component relates to emotional processing (output) [6, 14, 15].

Despite the importance of body-image disturbance, little is known about how the two body-image disturbance components are related to various eating-disorder characteristics. No previous study has investigated the associations between the two components and eating-disorder characteristics; as the two relate conceptually to different mental activities, they may be differentially associated with different eating-disorder characteristics. The affective component is expected to be associated with body-image concerns [10, 16]. Understanding differences in the associations between eating-disorder characteristics and the two body-image-disturbance components would provide insight into the psychological mechanisms underlying eating-disorder characteristics, the mechanisms underlying eating-disorder symptoms, and preventive and treatment strategies for eating disorders.

Perceived–actual and perceived–ideal discrepancies are suitable psychometric measures for the two components of body-image disturbance. The perceptual component is measured according to the discrepancy between the perceived and actual body size ("perceived–actual discrepancy") [12, 17]. Overestimation of body size underlies the perceptual component, and perceived–actual discrepancy best correlates with the perceptual component. The affective component can be quantified in various ways, including via self-reported questionnaires about feelings toward body image [10, 16], as well by the degree of discrepancy between the actual and ideal body size [18], and between the perceived and ideal body size [12, 14]. Most recent studies measured the affective component based on the discrepancy between the perceived and ideal body ("perceived–ideal discrepancy"). This measure of the affective component has the advantage of being directly based on the body image, unlike self-reported questionnaires investigating feelings toward the body image.

The Eating Disorder Inventory (EDI) [19] is frequently used to evaluate eating-disorder characteristics, and has been translated to several languages including Japanese. The structure of the Japanese version of the EDI2 (the most recent Japanese version) differs from that of the original version [20, 21]. The Japanese version of the EDI2 is composed of the following six factors: abnormal behavior related to eating (factor 1), dissatisfaction with one's own body (factor 2), negative self-evaluation (factor 3), maturity fear and confusion of the mind (factor 4), avoidance of interactions with others (factor 5), and desire for achievement (factor 6).

One previous study suggested that eating-disorder characteristics related to ineffectiveness and concerns about weight and shape were associated with each component of body-image disturbance. That study investigated the correlations of eating-disorder characteristics with perceived–actual and "actual–ideal" discrepancies, and showed that perceived–actual discrepancy correlated with ineffectiveness (a subscale of the EDI that includes items such as 'I cannot do anything at all') in eating-disorder patients [18]. Actual–ideal discrepancy, but not perceived–ideal discrepancy, was correlated with body dissatisfaction and drive for thinness (i.e., concerns about body weight and shape, exemplified by the EDI item 'My hips are too big') in both patients and healthy individuals.

The seminal study referred to above [18] was conducted around a quarter of a century ago; given recent advances in research techniques, the results seem to require updating in four respects: experimental design, conceptualization and measurement of disturbance, the version of the eating-disorder characteristics questionnaire used, and various confounding factors. First, the experimental design may have been susceptible to bias. The study participants adjusted images of their body themselves to match the ideal image on a life-sized screen. Therefore, participants could easily report that a silhouette image of a body corresponded to their own perceived body image, even if that was not the case. An anchoring effect may have been present [22]. Individuals who initially view an image of an extremely overweight body will tend to perceive their own body as being larger [23]. Second, the study did not investigate the perceived–ideal discrepancy as a measure of the affective component [18]. Rather, it investigated the relationships between the "actual–ideal" discrepancy and eating-disorder characteristics. However, the perceived–ideal discrepancy is now typically used as a measure of the affective component [12, 14]. The previous study reported a significant correlation between "actual–ideal" discrepancy and body dissatisfaction (EDI subscale score) in healthy controls, but not patients. However, there may be a significant correlation between "perceived–ideal" discrepancy and body dissatisfaction in eating-disorder patients. Third, as an older work, the previous study used the original EDI; newer versions are now available, i.e., the EDI2 and 3 [18, 21, 24], which include an additional 27 items (three subscales). Furthermore, the previous study did not observe any correlations between perceived–actual discrepancy and eating-disorder characteristics in healthy people. It is possible that a correlation would be detected using the newer versions of the EDI. Finally, potential confounders, such as body mass index (BMI), were not considered. BMI is associated with both perceived–actual and perceived–ideal discrepancies [12, 25, 26]. Healthy people with a higher BMI are more prone to body dissatisfaction [27–29]. Therefore, BMI could have influenced the previous study's results, in addition to the menstrual cycle and body checking. Several previous studies suggested changes in body size estimation and satisfaction in female patients before and during menstruation [30, 31]. Some people use mirrors to check body size daily, whereas others do not; this could influence the accuracy of body size estimations.

The present study aimed to determine the relationships between two body-image-disturbance components (perceptual and affective components) and eating-disorder characteristics represented by EDI2 scores. We predicted that ineffectiveness and concerns about body weight and shape (i.e., body dissatisfaction and drive for thinness) would be associated with perceived–actual and perceived–ideal discrepancies, respectively. To investigate this, we designed a forced-choice experiment based on the method of constant stimuli, in which participants viewed distorted silhouette images of their own photograph presented in a random order. We used a psychophysiological technique to reduce bias, such as social desirability bias [32], which affects responses on self-administered questionnaires. The participants were asked whether the body in the image was more overweight than their own or ideal body. Because the images were presented in a random order, the participants could not premeditate their answer [22].

Moreover, the method of constant stimuli prevented the anchoring effect compared with other psychophysiological experiments. Second, we used widely accepted measures of body-image disturbance: perceived–actual and perceived–ideal discrepancies. Third, we used the latest version of the Japanese EDI2 to evaluate eating-disorder characteristics (the EDI3 was not available in Japanese at the time of the experiments) [20, 21]. Finally, we controlled for possible confounding factors (BMI, age, menstrual cycle, and methods of body checking).

We hypothesized that perceived–actual discrepancy correlates with factors 3, 4, and 6 because these factors include items originally classified as ineffectiveness [20, 21]. Meanwhile, perceived–ideal discrepancy correlates with factors 1 and 2 because these factors include items originally classified as body dissatisfaction and drive for thinness [20, 21].

## 2 Method

### 2.1 Ethical approval

This study was approved by the Ethics Committee of Tohoku University Graduate School of Medicine, Japan. We obtained informed consent from the participants based on the principles of the Declaration of Helsinki.

### 2.2 Participants

Thirty healthy young adult women (mean age = 21.0 ± 1.38 years) were recruited. All participants were Japanese undergraduates or graduates of Tohoku University. It has been reported that some university students have eating-disorder characteristics and body-image disturbance [11, 12, 16]. Many similar previous studies recruited healthy young adult women, including university students. Therefore, the study participants of this study were comparable with those of previous studies. Participants were invited to participate in this study via an advertisement displayed in the university office. The participants' characteristics are shown in Table 1.

A sample size analysis was conducted using G*Power 3.1 [33]. Based on the correlation coefficients reported in previous studies [18, 34] between body-image disturbance and psychopathological questionnaire scores, such as the EDI2, the estimated effect size (effect size = 0.5) was determined. Assuming a two-tailed significance level of 0.05, an effect size of 0.5, and a power of 0.8, the required sample size for our study was 29. Considering the risk in the current study (wherein participants considered their own and ideal body images), a minimal number of participants was recruited.

Mean age, BMI, and Eating Disorder Inventory 2 (EDI2) scores of 30 young women. Both the transformed (calculating raw ratings for 1, 2, and 3 as zero and using only raw ratings for

**Table 1. Characteristics of the study participants.**

| | Mean (±SD) | | |
|---|---|---|---|
| Age | 21.0 (±1.38) | | |
| BMI (kg/m$^2$) | 20.7 (±2.10) | | |
| EDI2 | Transformed scoring | Non-transformed scoring | |
| Total | 53.1 (±19.6) | 227 (±32.1) | Cronbach α |
| Factor 1 Abnormal behavior related to eating | 6.93 (±5.06) | 34.4 (±10.3) | 0.85 |
| Factor 2 Dissatisfaction with one's body | 16.2 (±8.68) | 45.9 (±12.6) | 0.92 |
| Factor 3 Negative self-evaluation | 6.60 (±7.62) | 39.1 (±13.2) | 0.9 |
| Factor 4 Maturity fear and confusion of the mind | 5.73 (±3.49) | 29.7 (±5.21) | 0.68 |
| Factor 5 Avoidance of interactions with others | 10.3 (±6.51) | 48.2 (±9.75) | 0.87 |
| Factor 6 Desire for achievement | 8.17 (±4.36) | 27.5 (±5.64) | 0.8 |

5, and 6) and non-transformed scores (raw ratings without transformation) are shown. The Cronbach α was also calculated based on the non-transformed scores.

One participant was excluded from the correlation analysis between body-image disturbance components and EDI2 factors owing to a lack of data. Another was excluded from the correlation analysis involving perceived–ideal discrepancy; this was because we could not measure her perceived–ideal discrepancy owing to poor performance in the behavioral task.

## 2.3 Procedure

Participants attended our laboratory on two separate days. On day 1, we took photographs of the participants, which we changed into silhouette images. The participants underwent a psychological experiment using their own silhouette images as stimuli on day 2. This experiment comprised a task in which the participants judged their actual bodies, and one in which they judged their ideal bodies.

**2.3.1 Day 1: Preparation of stimuli.** First, we photographed the participants' entire bodies using a digital camera (PENTAX K-S2; RICOH, Tokyo, Japan). The participants stood upright, with their feet shoulder-width apart with their back against a white wall. All participants' postures and clothes (sports brassiere and leggings; S1 Fig) were identical. Silhouette images of each participant's whole body, excluding their heads, were created from their digital photograph using Adobe Photoshop CC 2017. Twenty-one images were created for the actual-body task stimuli by manipulating the width from −10% to +10% of the original width in steps of 1%. For the ideal-body task stimuli, 35 images were created by manipulating width: 31 images from −15% to +15% of the original width with steps of 1%; and four images distorted by −25%, −20%, +20%, and +25% relative to the original image. The range of distortion in each task was determined based on a preliminary experiment (n = 20; all participants were female undergraduate and graduate students of Tohoku University; mean age = 20.89 ± 1.29 years), so it covered the typical range of distortion in our population.

**2.3.2 Day 2: Day of the experiment.** After completing the EDI2 questionnaire, participants performed the actual-body and ideal-body tasks (Fig 1). The participants were informed in advance that they would be shown silhouette images of their body distorted only in terms of width, and that they would be required to estimate their actual and ideal body size according to the width of the images. In each trial, the participant was presented with a modified silhouette image of their body, and they judged whether the presented body image was wider than their actual body image in the actual-body task (Fig 1A; "Is the width of the silhouette larger than your actual body?") and wider than their ideal body image in the ideal-body task (Fig 1B; "Is the width of the silhouette larger than your ideal image?"). Participants were instructed to answer while they were looking at their silhouette.

Participants looked at the silhouette image for 3 sec and pressed the key button within 1 sec. The interval of the trial was 2 sec and the fixation image was presented during the interval. Participants looked at the set of stimuli (21 images from −10% to +10% or 35 images from −25% to +25%) three times in random order; thus, the actual-body task had 63 trials and the ideal-body task had 105 trials.

The participants' silhouette images were displayed on an 18.1-inch monitor (FlexScan L676 EIZO) in front of the participants. The participants put their chin on a chin rest to fix the position between the participant and the monitor. The monitor was located in the central 20˚ of the participants' visual fields, and silhouette images were located within the central 10˚ of participants' visual fields. These tasks were controlled using PsychoPy2 v1.83.0 [35, 36].

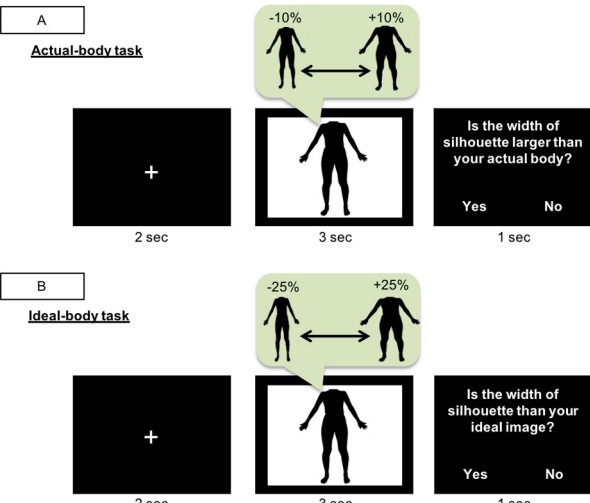

**Fig 1. Design of the actual-body and ideal-body tasks.** In each trial, participants looked at their own distorted silhouette image for 3 s and then answered a question by pressing the appropriate key. (A) In the actual-body task, participants answered "Yes" if the width of the image exceeded that of their actual body and "No" if it did not. (B) In the ideal-body task, participants answered "Yes" if the width of the image exceeded that of their ideal body and "No" if it did not.

## 2.4 Measures

### 2.4.1 Calculation of perceived–actual and perceived–ideal discrepancies.
In accordance with earlier studies [12, 14, 37], we identified perceived–actual and perceived–ideal discrepancies, which were operationalized above. The participants' perceived self-image and ideal-body-image sizes were estimated based on their choices in the actual- and ideal-body tasks, respectively (Fig 2). Using logistic regression, the participants' choice behavior was modeled as $P_{yes} = 1/[\exp{(-\beta X - \alpha)} + 1]$, where $P_{yes}$ denotes the probability of yes, $X$ indicates the degree of

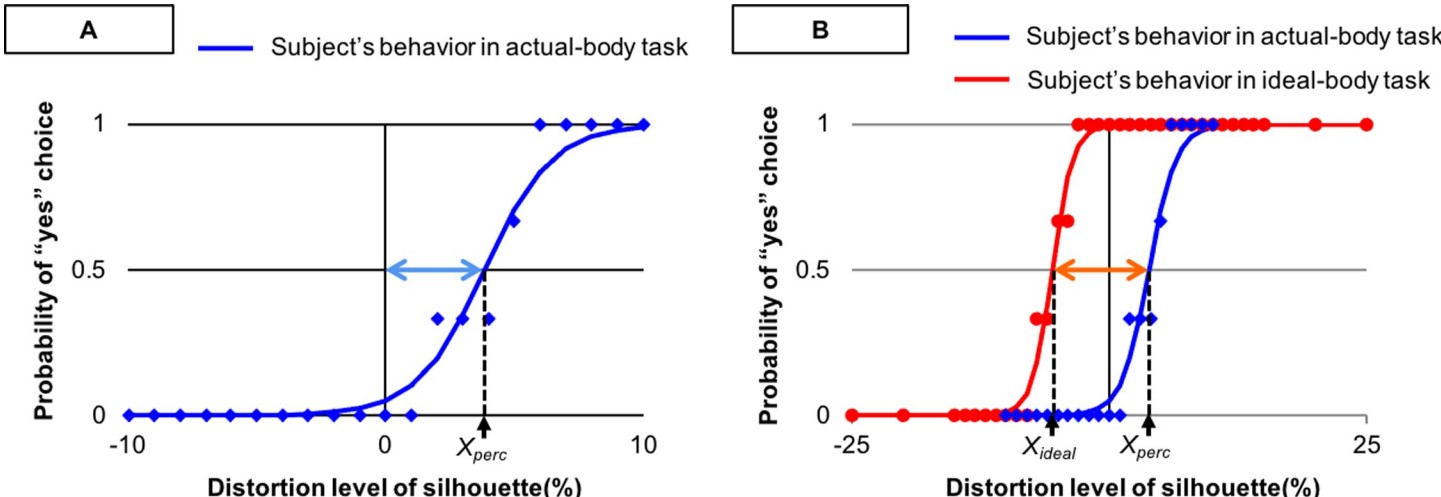

**Fig 2. Procedure for estimating perceived–actual discrepancy and perceived–ideal discrepancy.** A representative example of a participant's behavior during the actual-body (blue line) and ideal-body (red line) tasks according to logistic regression modeling. (A) The right blue arrow represents the perceived–actual discrepancy, defined as a probability of choosing 'Yes' on the actual-body task of 0.5 ($X_{perc}$). (B) The orange arrow represents the perceived–ideal discrepancy, defined as the discrepancy between a probability of choosing 'Yes' on the actual-body task of 0.5 ($X_{perc}$) and that on the ideal-body task ($X_{ideal}$).

distortion of the silhouette, and the free parameters α and β govern the shape of the slope (see the blue and red lines in Fig 2). Notably, the distortion level $X$ at which the participants changed their choice from 'yes' to 'no' corresponds to the size of the perceived or ideal body, represented as $X_{perc}$ (Fig 2A) and $X_{ideal}$ (Fig 2B), respectively.

Next, we quantified the perceived–actual and perceived–ideal discrepancies. Perceived–actual discrepancy was obtained by subtracting the participant's actual body from their perceived body ($X_{perc}$–zero; blue arrow in Fig 2A). Perceived–ideal discrepancy was obtained by subtracting the participant's ideal body from their perceived body ($X_{perc}$–$X_{ideal}$; orange arrow in Fig 2B). A positive perceived–actual discrepancy value indicated that the participant's estimate of their own body was larger than their actual body. For example, if a participant's perceived–actual discrepancy was 5, the participant overestimated her body size by 5%. A positive perceived–ideal discrepancy value indicated that the participant believed her body to be larger than their ideal body. Similarly, if a participant's perceived–ideal discrepancy was 5, the participant wishes to be 5% thinner than her perceived self-image.

**2.4.2 Questionnaire.** The participants completed the EDI2 [20] before performing the actual-body and ideal-body tasks. The questionnaire had 68 items: factors 1–6 had 13, 11, 14, 9, 14, and 7 items, respectively. Participants answered each question, such as 'I eat when I am upset,' according to a six-point scale from 1 = never to 6 = always. The EDI2 evaluates eating-disorder characteristics in healthy individuals. The EDI2 scores for healthy individuals without any eating-disorder symptoms were significantly lower compared to healthy individuals with eating-disorder symptoms [38]. The present study did not adopt the modified scoring system that is usually used for screening or evaluating symptom severity in patients because it has been suggested that raw scores (i.e., we scored '1' if participants rated '1') are better when participants are healthy [39]. The results are shown in Table 1.

**2.4.3 Body mass index.** Participants' height and weight were measured after taking their photograph, and the BMI was then calculated.

**2.4.4 Menstrual cycle.** Participants were divided into two groups: within 1 week before menstruation or mid-menstruation, and in other parts of the menstrual cycle. In the correlation analysis, dummy variables were used and the effect of menstruation on body-image disturbance was included as a covariate.

**2.4.5 Daily body-checking behavior.** Participants indicated which means of body checking they considered most important from among several options, such as measuring weight, checking their image in a mirror, or measuring body-fat percentage. The participants were divided into two groups: those who used mirrors as the most important means of checking their body and those who did not use mirrors as the most important means of evaluating their body. Dummy variables were used in the correlation analysis and familiarity with looking at their whole body was included as a covariate.

## 2.5 Data analysis

Data were analyzed using R version 3.3.2 (R Core Team 2014. R: A language and environment for statistical computing. R Foundation for Statistical Computing, Vienna, Austria, http://www.R-project.org/).

To determine whether the perceptual and affective components were related to different eating-disorder characteristics, we conducted a partial correlation analyses. We performed a partial correlation analysis between the perceived–actual discrepancy and the six EDI2 factors, controlling for the participant's perceived–ideal discrepancy, BMI, age, menstrual cycle, type of daily body-checking behavior, and the interval between days 1 and 2. Similarly, a partial correlation analysis was performed between the perceived–ideal discrepancy and the six EDI2

factors controlling for perceived–actual discrepancy, BMI, age, menstrual cycle, type of daily body-checking behavior and the interval between day 1 and 2. All correlation analyses employed Pearson's correlation coefficients.

Bonferroni correction was applied for the partial correlation analyses. The correlations of perceived–actual discrepancy with factors 3, 4, and 6, and of perceived–ideal discrepancy with factors 1 and 2, were analyzed. Therefore, to calculate Bonferroni-corrected p-value, we multiplied the uncorrected p-value for each factor partially correlated with perceived–actual discrepancy and perceived–ideal discrepancy by 3 and 2, respectively. Bootstrapping analysis using 5,000 bootstrap samples was also conducted. When the 95% confidence intervals do not include zero, the partial correlation coefficient can be regarded as significant.

The correlation between perceived–actual discrepancy and perceived–ideal discrepancy was also investigated. If each component of body-image disturbance is associated with different aspects of eating disorders, there should be no significant correlation between perceived–actual discrepancy and perceived–ideal discrepancy.

## 3 Results

### 3.1 Descriptive data of the two body-image disturbance components

Participants estimated their bodies as being significantly larger than their bodies actually were (perceived–actual discrepancy = 4.14 ± 3.22, t(29) = 7.04, p < 0.001; Table 2) and their estimated self-body image was significantly larger than their ideal body image (perceived–ideal discrepancy = 5.78 ± 7.51, t(28) = 4.14, p < 0.001; Table 2). These results suggest that, on average, participants perceived their body as 4.14% larger than their actual body, and their ideal body size was 5.78% smaller than their perceived body size. The across-participants correlation between perceived–actual discrepancy and perceived–ideal discrepancy was not significant (r [27] = 0.10, p = 0.60); a lack of correlation between perceived–actual discrepancy and perceived–ideal discrepancy was observed in the scatterplot (Fig 3).

### 3.2 Partial correlations between each body-image disturbance component and the EDI2 factors

Consistent with the present hypotheses, there was a partial correlation between perceived–actual discrepancy and the negative self-evaluation score, i.e., factor 3 of the EDI2 (r[27] = 0.51, Bonferroni-corrected p-value = 0.039; Fig 4A). The 95% confidence intervals did not include zero (CI = [0.042, 0.812]), indicating that the partial correlation was significant. Perceived–ideal discrepancy showed a partial correlation with the score for dissatisfaction with one's own body, i.e., factor 2 of the EDI2 (r[26] = 0.52, Bonferroni-corrected p-value = 0.026; Fig 4B). Again, significance was indicated by the 95% confidence intervals not including zero (CI = [0.041, 0.794]). These results were obtained after controlling for potential confounders such as BMI, age, menstrual cycle, daily body-checking behavior, and the interval between

**Table 2. Descriptive statistics for body-image disturbance.**

|  | Mean (±SD) | Min | Max | 1st Qu | Median | 3rd Qu |
|---|---|---|---|---|---|---|
| Perceived-actual discrepancy (%) | 4.14 (±3.22) | -2.43 | 12.7 | 2.50 | 4.21 | 6.20 |
| Perceived-ideal discrepancy (%) | 5.78 (±7.51) | -12.3 | 22.3 | 1.01 | 5.90 | 11.2 |

Descriptive statistics for perceived–actual discrepancy and perceived–ideal discrepancy. The means of perceived–actual discrepancy and perceived–ideal discrepancy were both greater than zero (one-sample t-test, df = 29, p < 0.001).

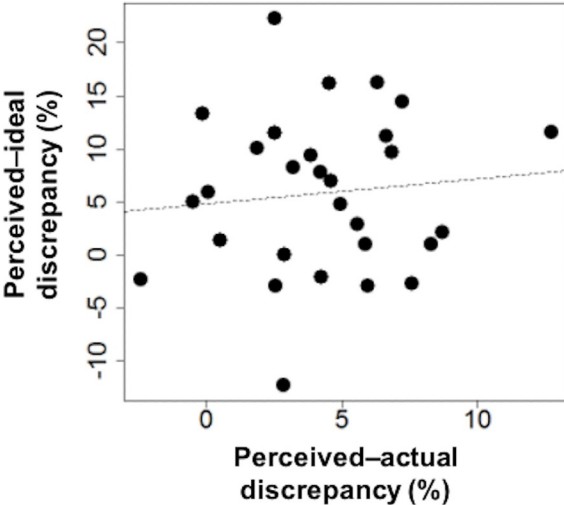

**Fig 3. Scatterplot of the perceived–actual discrepancy and perceived–ideal discrepancy.** The correlation between perceived–actual discrepancy and perceived–ideal discrepancy was not significant (df = 27, r = 0.10, p = 0.60).

days 1 and 2 (see Table 3 for the results of the partial correlation analysis). These results are unlikely to be due to outliers (Fig 4).

## 4 Discussion

The purpose of this study was to determine whether each of the two body-image-disturbance components were associated with distinct eating-disorder characteristics in healthy individuals. A psychophysical method, i.e., the method of constant stimuli, was used to determine participants' perceived–actual and perceived–ideal discrepancies based on their judgments about their silhouette images. Then, partial correlation analyses between each body-image

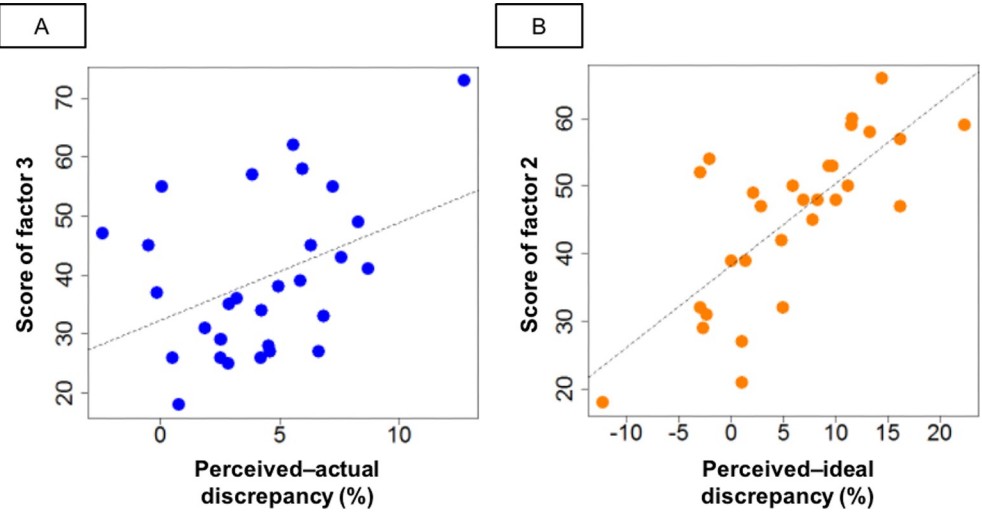

**Fig 4. Scatterplot between perceived–actual discrepancy and factor 3, and of perceived–ideal discrepancy and factor 2.** Simple correlation analyses between perceived–actual discrepancy and factor 3 (A), and of perceived–ideal discrepancy and factor 2 (B), were performed for illustrative purposes, while the results of the partial correlation analysis were subjected to further statistical tests. The correlation coefficients were r = 0.51 (p = 0.013) and r = 0.52 (p = 0.013), respectively. Both correlations remained significant after Bonferroni correction.

Table 3. Relationships between body-image disturbance components and EDI2 factors, after controlling for potential confounders (e.g., BMI).

| | Correlation coefficient | t-value | p-value |
|---|---|---|---|
| **Partial correlation analysis** | | | |
| Perceived–actual discrepancy (n = 29) | | | |
| Factor 1 Abnormal behavior related to eating | 0.09 | 0.41 | 0.68 |
| Factor 2 Dissatisfaction with one's body | -0.09 | 0.41 | 0.68 |
| **Factor 3 Negative self-evaluation** | **0.51** | **2.72** | **0.013***  |
| **Factor 4 Maturity fear and confusion of the mind** | **0.27** | **1.29** | **0.21** |
| Factor 5 Avoidance of interactions with others | -0.13 | 0.60 | 0.55 |
| **Factor 6 Desire for achievement** | **0.08** | **0.37** | **0.72** |
| Perceived–ideal discrepancy (n = 28) | | | |
| **Factor 1 Abnormal behavior related to eating** | **0.36** | **1.73** | **0.10** |
| **Factor 2 Dissatisfaction with one's body** | **0.52** | **2.72** | **0.013***  |
| Factor 3 Negative self-evaluation | 0.02 | 0.089 | 0.93 |
| Factor 4 Maturity fear and confusion of the mind | -0.06 | 0.27 | 0.79 |
| Factor 5 Avoidance of interactions with others | -0.1 | 0.45 | 0.66 |
| Factor 6 Desire for achievement | 0.14 | 0.63 | 0.53 |

Partial correlation analysis of perceived–actual and perceived–ideal discrepancies with EDI2 factors. BMI, age, menstrual cycle, type of body-checking behavior (use of a mirror or not), and the interval between day 1 (when a participant's photograph was taken) and day 2 (the experimental day) were controlled for. The correlations with the a-priori hypothesis are displayed in bold.

* Bonferroni-corrected $p < 0.05$

disturbance component and eating-disorder characteristics were conducted. We used the Japanese version of the EDI2, which is the latest version in translation, and controlled for several potential confounders, including BMI, in the analyses. This study is the first to report that the two components of body-image disturbance were associated with different EDI2 factors: perceived–actual discrepancy was correlated with EDI2 factor 3, which is related to negative self-evaluation, whereas perceived–ideal discrepancy was correlated with EDI2 factor 2, which is related to dissatisfaction with one's own body. These results support the notion that the two components of body-image disturbance are independent [6, 14].

The observed positive correlation between perceived–actual discrepancy and negative self-evaluation (factor 3) supports a previous hypothesis that attentional bias underlies perceived–actual discrepancy. We showed that, among the three EDI2 domains of ineffectiveness, perceived–actual discrepancy was related to negative self-evaluation rather than maturity fear and confusion of the mind (factor 4) or desire for achievement (factor 6). Previous studies reported that attentional bias toward the waist was associated with perceived–actual discrepancy in a healthy population [13, 17]. In particular, healthy people with greater perceived–actual discrepancy exhibited attentional bias toward one side of the torso [17]. Previous studies also reported a relationship between negative self-evaluation and attentional bias. Low self-esteem has been proposed as one of the reasons for attentional bias toward disliked parts of one's own body [40]. We observed a relationship between perceived–actual discrepancy and negative self-evaluation, which bridges the results of previous studies and supports the hypothesis that attentional bias underlies perceived–actual discrepancy. Individuals who exhibit negative self-evaluation would tend to look at body parts that they are dissatisfied with, such as the waist; this gaze pattern may contribute to perceived–actual discrepancy. Factors 4 and 6 rarely include items related to negative self-evaluation. Thus, these factors may not relate to attentional bias, which could explain the lack of significant correlations with perceived–actual

discrepancy. This study implies an involvement of attentional bias; future studies should further explore this possibility.

The positive correlation between perceived–ideal discrepancy and dissatisfaction toward one's body (factor 2) may reflect body dissatisfaction independent from one's actual body size. The partial correlation analysis revealed that only dissatisfaction toward one's body was correlated with perceived–ideal discrepancy after controlling for confounding factors such as BMI. Overweight people tend to be dissatisfied with their bodies [27–29]. We also observed that BMI was strongly correlated with dissatisfaction toward one's body (r[27] = 0.71, t = 5.22, p < 0.001) and perceived–ideal discrepancy (r[26] = 0.61, t = 3.94, p < 0.001). However, the correlation between perceived–ideal discrepancy and dissatisfaction toward one's body remained significant after controlling for BMI. Irrespective of their actual body shape and weight, participants with a greater perceived–ideal discrepancy were less satisfied with their body. This suggests that the perceived–ideal discrepancy reflects body dissatisfaction independent of actual body weight and shape.

Based on the above, body dissatisfaction independent of actual body weight and shape may result from a deficit in a feedback circuit between change in body dissatisfaction and change in actual body weight and size. Most, but not all, overweight people report less dissatisfaction with their body after weight loss. Body dissatisfaction independent of the actual body weight and size may explain why some people persist with a diet. This is consistent with a previous study that reported that some healthy people with higher body dissatisfaction and a low BMI continue to diet [3]. In contrast to the relationship between perceived–ideal discrepancy and factor 2, the correlation between perceived–ideal discrepancy and factor 1 was not significant after controlling for BMI, which suggests that this correlation was strongly influenced by participants' actual body weight and shape. Another possible reason for the lack of a significant correlation between perceived–ideal discrepancy and factor 1 is that factor 1 also includes items related to bulimia (except for those related to body weight and shape). A longitudinal study is needed to confirm whether the perceived–ideal discrepancy reflects refractory body dissatisfaction independent of actual body weight and shape change.

The correlation between perceived–actual discrepancy and perceived–ideal discrepancy was not significant in this study, consistent with the notion that these two body-image disturbance components are independent. Specifically, perceived–actual and perceived–ideal discrepancies were correlated with different EDI2 factors, and were only weakly correlated with each other. The apparent independence of these two factors is consistent with the findings of previous neuroimaging studies that investigated task-related brain activity [6, 14, 15]. In particular, one study reported that the brain activity observed during a task requiring estimation of one's actual body size differed from that observed during a task requiring estimation of one's ideal body size [14].

In contrast to the present study's findings, several previous studies suggested that perceived–actual discrepancy is correlated with questionnaire scores on concerns regarding body weight and shape [18, 41]. This inconsistency may be attributable to the following two factors. First, in contrast to previous studies, the present study controlled for confounding factors, such as BMI. Second, these earlier studies recruited patients, who showed a correlation between perceived–actual discrepancy and concerns regarding body weight and shape. This apparent difference between patients and healthy people may promote speculation about the mechanisms underlying the development of eating disorders. The differences between patients and healthy individuals might explain the undue influence of weight and shape on self-esteem in some individuals. This study hypothesized that the two body-image disturbance components would be independent of each other in healthy people, but not in those at risk of an eating disorder. In the latter population, negative self-evaluation (i.e., perceived–actual

discrepancy) may be associated with dissatisfaction toward one's own body (i.e., perceived–ideal discrepancy). In such individuals, weight and shape may exert an undue influence on self-esteem, ultimately leading to eating disorders. This highlights the importance of performing basic studies that include both healthy people and patients to better understand the mechanisms underlying eating disorders.

Our results supported a relationship between perceived–actual discrepancy and attentional bias, which may be clinically relevant since it is in line with interventions suggested in recent studies. One study reported that a full-body illusion created using virtual-reality techniques improved perceived–actual discrepancy [42]. Another study reported that a virtual reality-based body exposure therapy reduced attentional bias [43]. Decreasing attentional bias may improve perceived–actual discrepancy, and virtual-reality techniques may be one of the most effective ways to decrease attentional bias toward particular body parts. An interventional study is needed to investigate the relationship between improved perceived–actual discrepancy and attentional bias.

This study had several limitations. First, only healthy women were recruited so it is difficult to compare the results with studies that included patients with eating disorders. Although similarities between healthy people and patients have been reported [38, 44–46], patient studies will be necessary to validate the relationships found in this study. Second, our results have limited cross-national generalizability. The BMI of the participants in the current study was similar to that in previous body-image studies conducted in Japan [47, 48], but lower than that in studies conducted in Western countries [12, 13]. The lack of obese individuals (BMI > 25.0 kg/m$^2$) in the current study is explained by the low prevalence of obesity in Japanese females in their twenties (i.e., 8.9% according to a national survey conducted in 2019 by the Ministry of Health, Labor and Welfare). BMI is a critical factor in body-image studies because it is strongly correlated with concerns about body image [12, 27–29]. The comparability of our findings with those of studies that recruited participants from different cultures is also limited by our use of the Japanese version of the EDI2, which has a culture-specific six-factor structure. However, the results for factor 2 may be directly comparable to those of other studies because this factor is primarily composed of body-dissatisfaction items from the original EDI2 subscale. In contrast, factor 3 is composed of items from four subscales of the original EDI2: ineffectiveness, interoceptive awareness, impulse regulation, and social insecurity. In addition, this study had a small sample, although it was based on a power analysis. Due to the small sample size, relationships between the two components of body-image disturbance and eating disorder characteristics with small effect sizes may have been missed. Biological and cultural factors reflected in the lower BMI range and structure of the EDI2 would have influenced our findings and limit the generalizability of the current study. International comparative studies including larger samples and wider ranges of BMI will be needed to confirm our results.

## 5 Conclusion

In conclusion, this study demonstrated that perceptual and affective components were associated with different eating-disorder characteristics. Specifically, perceived–actual discrepancy was related to negative self-evaluation, whereas perceived–ideal discrepancy was related to dissatisfaction with one's own body. Together with data showing the lack of a significant correlation between perceived–actual discrepancy and perceived–ideal discrepancy, these results indicate that, in healthy populations, the two body-image disturbance components are independent of each other. Given that the correlation between perceived–ideal discrepancy and dissatisfaction with one's own body remained significant after controlling for BMI, perceived–ideal discrepancy could be an indicator of body dissatisfaction independent of the actual body

weight and shape. In summary, our results confirmed the relationship between body-image disturbances and eating-disorder characteristics in healthy women, and thus provide new insight into the psychological mechanisms related to eating-disorder characteristics.

## Supporting information

**S1 Table. Participants data.**
(XLSX)

**S1 Fig. Clothing worn by participants when photographs were taken.** Participants were loaned a sports brassiere and leggings according to their height. If the clothes were not comfortable, we provided a medium-large-sized garment (the standard size was small-medium). (TIF)

## Author Contributions

**Conceptualization:** Yumi Hamamoto.

**Formal analysis:** Yumi Hamamoto.

**Funding acquisition:** Yumi Hamamoto, Shinsuke Suzuki, Motoaki Sugiura.

**Investigation:** Yumi Hamamoto.

**Methodology:** Yumi Hamamoto, Shinsuke Suzuki, Motoaki Sugiura.

**Project administration:** Yumi Hamamoto.

**Resources:** Shinsuke Suzuki, Motoaki Sugiura.

**Supervision:** Shinsuke Suzuki, Motoaki Sugiura.

**Visualization:** Yumi Hamamoto.

**Writing – original draft:** Yumi Hamamoto.

**Writing – review & editing:** Yumi Hamamoto, Shinsuke Suzuki, Motoaki Sugiura.

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
