## [Decision Letter · Decision Letter 0]

17 Jun 2021

PONE-D-21-08191

Two components of body-image disturbance are differentially associated with distinct aspects of eating-disorder tendencies in healthy young women

PLOS ONE

Dear Dr. Hamamoto,

Thank you for submitting your manuscript to PLOS ONE. After careful consideration, we feel that it has merit but does not fully meet PLOS ONE’s publication criteria as it currently stands. Therefore, we invite you to submit a revised version of the manuscript that carefully and systematically addresses ALL the points raised by both reviewers during the review process (see below).

We look forward to receiving your revised manuscript.

Kind regards,

Emmanuel Manalo, PhD

Academic Editor

PLOS ONE

Journal Requirements:

Additional Editor Comments (if provided):

Both reviewers acknowledged the potential contribution that your research can make to this research area. At the same time, they provided detailed comments about the revisions that you need to make to bring the manuscript up to a level that can be considered appropriate for publication. So, could you please systematically go through all the points that these reviewers have made and carefully address those in revising your manuscript? If you have any questions at all, please do not hesitate to contact me or the PLOS ONE staff.

Reviewers' comments:

Reviewer's Responses to Questions

**Comments to the Author**

1. Is the manuscript technically sound, and do the data support the conclusions?

Reviewer #1: Yes

Reviewer #2: Partly

2. Has the statistical analysis been performed appropriately and rigorously? 

Reviewer #1: Yes

Reviewer #2: Yes

3. Have the authors made all data underlying the findings in their manuscript fully available?

Reviewer #1: Yes

Reviewer #2: Yes

4. Is the manuscript presented in an intelligible fashion and written in standard English?

Reviewer #1: No

Reviewer #2: Yes

5. Review Comments to the Author

Reviewer #1: Thank you for providing me with an opportunity to review this manuscript. Strengths of the manuscript include the design and experimental procedure used. Moreover the data was analyzed appropriately which included controlling for a number of possible confounding variables. My decision of "Major Revisions" rather than "Reject" was based on these strengths. However, I found reading this manuscript very difficult to read primarily because of the very poor clarity of expression especially in the Introduction and the Discussion sections. Moreover, much of the studies cited in the manuscript are very dated. Before this manuscript can be accepted for publication the authors need to address the following:

Abstract: The abstract contains very little information about the sample used (i.e., who were the participants? mean age, SD, etc). This section needs some fine-tune editing (e.g., the first sentence is not necessary nor the the description of the study as a "well controlled psychophysical experiment"). It would be better if the authors provided a brief description of the procedure used to measure perceptual distortions (what is referred to here as "perceived-actual discrepancy) and the cognitive/affective component of body dissatisfaction (which is referred to here as "Perceived-ideal discrepancy'). Moreover, it is better to refer to the eating disorder characteristics/symptoms measured by the EDI-2 factors rather than naming the factors.

Introduction: My main difficulty in reading this section is in the description of the "perceived-actual discrepancy" and the "perceived-ideal discrepancy" lacks clarity and these terms are not consistent with those reported in the cited literature. Therefore, it was very difficult to marry up what is described in this section and what is described/reported in the cited studies. For example, is the "perceived-actual" discrepancy referring to body image distortions (i.e., a distortion in perception)? Is the "perceived-ideal discrepancy" referring to body dissatisfaction which is regarded in the literature as being an affective/attitudinal component of body image distortion? If this is the case then the authors should not refer to this component as a "motivational" component as this is very misleading. The authors need to clearly describe the difference between these two forms of body-image disturbance, e.g., in regards to the second component, the individual may be able to accurate perceive their size, shape etc but there is a discrepancy between their ideal body shape/size and their actual body shape/size.

Similarly, the authors refer to eating-disorder tendencies instead of eating disorder characteristics or symptoms. The inconsistencies in terminology used in this manuscript versus those in published literature combined with the poor clarity of expression makes reading this section and understanding the goals of the study very confusing and difficult.

The rationale for the study needs to be presented more strongly and clearly. The authors are basically replicating a very dated study (i.e., Probst et al., 1998) but using better methodology, measure of disordered eating and controlling for a wider range of possible confounding variables. But why is there a need to do this? Has there been contention/debate in the literature about how body distortion versus body dissatisfaction are associated with different disordered eating characteristics/symptoms?

Methodology: Rationale for recruiting University students needs to be provided. Moreover, why was the focus of judgment in the "Experiment" on the width of the body and not on the size of the body (i.e., “Is the width of the silhouette larger than your ideal image? versus "Is the silhouette larger than your ideal image?).

Results: The Cronbach's alpha of the EDI-2 factors/subscales for the current sample needs to be reported either in the Method or Result section. Otherwise, the data appears to be appropriately analyzed. However, the clarity of expression in this section can be improved. For example, why do we need the scatter plots? How do they assist in reporting the results?

Discussion: While the findings of this study do confirm the findings of previous studies on the relationship between body image disturbances and disordered eating characteristics/symptoms, it is not clear what new insights this study provides. This needs to be clearly explained. Suggestion for future research are also needed.

I can sympathize and understand the difficulties in writing a manuscript in a language different to one's first language. I recommend that the authors get their manuscript checked by readers who are both proficient in English and familiar with the relevant literature.

Reviewer #2: This manuscript was interesting to read, highlighting the need for researchers and clinicians to consider different aspects of body image. It uses a novel approach of measuring perceived body, and how aspects of body image are differentially related to factors in EDI2. As such, the manuscript have some important findings that should be of interest in the field of both eating disorders and body image. However, I believe that it needs some revisions and clarifications before it is ready for publication.

One main concern is the mode of measurement. In the field of body image, there is a plethora of different measurements, stemming from the multidimensional construct of body image. Different researchers focus on different aspects of body image, thus measuring different constructs. The present study seems to be using an experimental mode of measurement. The measurement seems innovative, and do need to be more thoroughly introduced to the reader in the introduction. Why did authors choose this approach? How does it relate to other similar measurements? Is this a validation study?

Another main concern regards the introduction, which I found a bit confusing. For instance, it is not clearly stated which of the cited references considers components of body image in patients with an eating disorder, and which are derived from a normal population. I found the same confusion in the term “eating-disorder tendencies”, is this part of eating disorder symptoms or disordered eating in the normal population? There is agreement between researchers that aspects of body image differ in individuals with or without an eating disorder. Thus, the introduction would benefit from separating research, or at least comment on the matter. Further, I have trouble with the choice of the term “motivation” as describing one of the body image constructs, which I am not familiar with. Could you provide a reference to this terminology? Or is the “affective” component you are referring to?

A third main concern regards the structure of the manuscript. In general, the manuscript is a bit confusing to read, where authors needs to rearrange some of the text. They are not following their own headings, mixing details between introduction and method. I will mention a few on the points below.

Following are some further considerations in the different parts of the manuscript:

• The article begins with describing that body image disturbance consists of two components and cites three studies supporting this claim. When reading the studies, they do include more components- perceptive, affective, cognitive and a behavioral one. Body image is a complex structure, and your manuscript could be more clear about why you have selected these two components, or at least suggest these two are not the only components suggested in the literature.

• On page 5, critique is raised against a specific study. I think this could be stated in a more humble way, pointing to limitations raised in general. For instance, the notion on using an outdated version of EDI, for a study published in -98, while the present article uses a version of EDI that would also be considered obsolete in many instances of contemporary research.

• Aim. The aim would benefit from a definition of the term “eating-disorder tendencies” in the introduction.

• The sentence following the aim seems to describe methodology, would you consider moving it for clarity? Thus, the stated hypothesizes (“we predicted”-change to hypothezised?) directly follows on the aim. Moreover, the description on the Japanese version of EDI2 would benefit of a move to earlier in the introduction, with a description on the factors identified and what you believe is important for your study. Further, since you have described that an EDI3 exists, why did you settle for EDI2?

• Page 7, the last two sentences in the introduction. These are very unclear, I do not understand what you are trying to say, please read these and consider rephrasing.

• Method. Under the heading of participants, you could consider the number and characteristics of the sample as part of your results. I could not find any descriptions on how participants were recruited, were they recommended participation, or did they respond to advertisement?

• Measures. In the description on quantification (page 13), adding possible ratings that participants could receive would aid in understanding your scale. Without this clarification, your results in Table 2 are hard to understand and interpret. For instance, how should I interpret the mean of 4.14? That the mean percentage of discrepancy is 4.14?

• 2.4.2 Questionnaire. EDI2 is described as distinguishing patients from controls-is it patients with an eating disorder, or another patient-group? Is there a cut-off score separating patients from healthy controls? Stating this will make it possible to judge if your study population is indeed healthy, at risk, or should even be considered as showing ratings indicative of an eating disorder. (again, this points as to why table 1 should be part of your results)

• 2.4.4. Menstrual cycle. In the section on BMI you describe how you measured it. I would suggest using this way of describing measurement in the method section, and moving descriptions and citations to either the introduction or discussion. Thus, first sentence in this heading would benefit from a removal.

• 2.4.5. Please see comment above.

• Results. 3.1. Mean levels reported needs to clarified, what are these numbers representing?

• 3.1. Here you state the correlation between perceived-actual and perceived-ideal (page 16), which is stated again on page 17?

• Table 3. Why are not all correlations between the different body image components and factors reported? The supporting information S1 Table uses abbreviations that are uninterpretable.

• Discussion. I found the discussion a bit confusing as well, what are the main findings in the current study?

• Page 20. The paragraph beginning in line 4. Here authors introduce attentional bias as an explanation. This is a bit speculative, and not part of the study results. Again, this paragraph lacks clarity, and as a reader I am confused as to how your results “implies that attentional bias plays a mediating role…”?

• Page 21, first sentence. Again, it is unclear, how does this “reflect an eating-disorder-related phenomenon”? You have not investigated patients with an eating disorder, nor reported any pathological tendencies in the study population.

• In the middle of page 21, one sentence starts with “This unreasonable body dissatisfaction”. Do you consider how a healthy individual perceives his/her body as target for judgment? Is there a wrong and right aspect to this? I think it is more humble to use your own terminology here, on discrepancy.

• Page 21, last sentences. Here you present results that have not been reported, and again, as a reader I am confused as to why only BMI is selected and not all factors you are controlling for. I don’t understand how this implies a mediation by BMI.

• page 23, the paragraph on the middle of the page. This sentence was confusing to read, this hypothesis was not part of the current study? Consider rephrasing.

• Conclusions. Again, is attentional bias part of the current study? Otherwise I don’t think it should be part of your conclusions.

6. PLOS authors have the option to publish the peer review history of their article (what does this mean?). If published, this will include your full peer review and any attached files.

Reviewer #1: No

Reviewer #2: **Yes: **Maria Fogelkvist

---

## [Author Response · Author response to Decision Letter 0]

31 Jul 2021

Dear the Editor of PLOS ONE,

Thank you for giving us the opportunity to submit our revised manuscript entitled “Two components of body-image disturbance are differentially associated with distinct eating disorder characteristics in healthy young women” to PLOS ONE. We appreciate your comments and those of the reviewers, and have revised the manuscript accordingly.

Our responses to the comments by the reviewers are as follows:

Response to Reviewer 1

General

Thank you for providing me with an opportunity to review this manuscript. Strengths of the manuscript include the design and experimental procedure used. Moreover the data was analyzed appropriately which included controlling for a number of possible confounding variables. My decision of "Major Revisions" rather than "Reject" was based on these strengths. However, I found reading this manuscript very difficult to read primarily because of the very poor clarity of expression especially in the Introduction and the Discussion sections. Moreover, much of the studies cited in the manuscript are very dated. Before this manuscript can be accepted for publication the authors need to address the following:

Reply

Thank you very much for reviewing our manuscript and providing insightful suggestions and comments. The reviewer raised four concerns regarding the terminology used in this study, the study rationale, and the language and references. Accordingly, we have revised the terminology (e.g., “motivational component” has been changed to “affective component”, as used in previous studies). Second, we have emphasized the rationale for the study, including by reference to the gaps in previous knowledge and importance of this topic. Third, we have had our manuscript reviewed by an expert on eating disorders and an English editing service. Finally, we have updated the reference list to include recent papers.

Comment 1

Abstract: The abstract contains very little information about the sample used (i.e., who were the participants? mean age, SD, etc).

Reply

Thank you very much for your comment. In accordance with the reviewer’s suggestion, we have added information about our sample size to the abstract. The text reads as follows:

“In a psychophysiological experiment, female university students (mean ± SD age = 21.0 ± 1.38 years) were presented with silhouette images of their own bodies that were distorted in terms of width.” (page 2, lines 29-30)

Comment 2

This section needs some fine-tune editing (e.g., the first sentence is not necessary nor the description of the study as a "well controlled psychophysical experiment"). It would be better if the authors provided a brief description of the procedure used to measure perceptual distortions (what is referred to here as "perceived-actual discrepancy) and the cognitive/affective component of body dissatisfaction (which is referred to here as "Perceived-ideal discrepancy').

Reply 

Thank you for your comment. In accordance with the reviewer’s suggestion, we have deleted the unnecessary first sentence and the description of the “well controlled psychophysical experiment”. We have added details of the psychophysical experiment used to measure perceived–actual and perceived–ideal discrepancies. The edited text reads as follows: 

“In a psychophysiological experiment, female university students (mean ± SD age = 21.0 ± 1.38 years) were presented with silhouette images of their own bodies that were distorted in terms of width. The participants were asked whether each silhouette image was more overweight than their actual or ideal body images.” (page 2, lines 29-32)

Comment 3

Moreover, it is better to refer to the eating disorder characteristics/symptoms measured by the EDI-2 factors rather than naming the factors.

Reply

Thank you for your suggestion. We agree that use of the original EDI2 subscale names (such as body dissatisfaction, bulimia, and perfectionism) would have made the text easier to understand. However, each factor subsumes several EDI2 subscales, especially factor 3. Factor 3 comprises ineffectiveness, interoceptive awareness, impulse regulation, and social insecurity. The Japanese version of the EDI2 is different from the original EDI2 in terms of subscale structure. Therefore, we preferred to use the factor names rather than eating-disorder characteristics/symptoms of the original EDI2. We have explained this in the Introduction as follows:

“The Eating Disorder Inventory (EDI) (19) is frequently used to evaluate eating-disorder characteristics, and has been translated to several languages including Japanese. The structure of the Japanese version of the EDI2 (the most recent Japanese version) differs from that of the original version (20, 21). The Japanese version of the EDI2 is composed of the following six factors: abnormal behavior related to eating (factor 1), dissatisfaction with one’s own body (factor 2), negative self-evaluation (factor 3), maturity fear and confusion of the mind (factor 4), avoidance of interactions with others (factor 5), and desire for achievement (factor 6).” (page 5, lines 98-106)

Comment 4

Introduction: My main difficulty in reading this section is in the description of the "perceived-actual discrepancy" and the "perceived-ideal discrepancy" lacks clarity and these terms are not consistent with those reported in the cited literature. Therefore, it was very difficult to marry up what is described in this section and what is described/reported in the cited studies. For example, is the "perceived-actual" discrepancy referring to body image distortions (i.e., a distortion in perception)? Is the "perceived-ideal discrepancy" referring to body dissatisfaction which is regarded in the literature as being an affective/attitudinal component of body image distortion? If this is the case then the authors should not refer to this component as a "motivational" component as this is very misleading. 

Reply

Thank you very much for your comment. We apologize that our terminology was inconsistent with that used in previous studies. The reviewer pointed out that the affective component should not be referred to as the “motivational component”, because it is misleading. We have revised the manuscript and now use the term “affective component”, which is consistent with the previous studies.

The reviewer raised concerns regarding the “perceived–actual” and “perceived–ideal” discrepancies. However, we would like to retain these terms in the current study to distinguish between the two components of body-image disturbance and the psychometric measurements thereof. The concept of two dissociable components of body-image disturbance (perceptual and affective) is relatively well-established. Individuals with the perceptual component overestimate their body size, while those with the affective component are dissatisfied with their body. Previous studies measured each component in different ways. For instance, some studies measured the affective component via self-reported eating-disorder questionnaires, while others measured it in psychological experiments. From a psychometric standpoint, the meaning of “affective component” differs according to the measurement method. Therefore, we believe that distinguishing between the two components and their psychometric measurements (perceived–actual and perceived–ideal discrepancies) is important. To clarify this, we have added the following text:

“Perceived–actual and perceived–ideal discrepancies are suitable psychometric measures for the two components of body-image disturbance. The perceptual component is measured according to the discrepancy between the perceived and actual body size (“perceived–actual discrepancy”) (12, 17). Overestimation of body size underlies the perceptual component, and perceived–actual discrepancy best correlates with the perceptual component. The affective component can be quantified in various ways, including via self-reported questionnaires about feelings toward body image (10, 16), as well by the degree of discrepancy between the actual and ideal body size (18), and between the perceived and ideal body size (12, 14). Most recent studies measured the affective component based on the discrepancy between the perceived and ideal body (“perceived–ideal discrepancy”). This measure of the affective component has the advantage of being directly based on the body image, unlike self-reported questionnaires investigating feelings toward the body image.” (pages 4-5, lines 84-97) 

Comment 5

The authors need to clearly describe the difference between these two forms of body-image disturbance, e.g., in regards to the second component, the individual may be able to accurate perceive their size, shape etc but there is a discrepancy between their ideal body shape/size and their actual body shape/size.

Reply

Thank you for your comment. In accordance with the reviewer’s suggestion, we have added a description of the two components of body-image disturbance. The perceptual component is related to overestimation of body size, and is associated with a discrepancy between the perceived and actual body image. The affective component is related to dissatisfaction and emotional problems related to body image, and is associated with a discrepancy between the ideal and perceived body size. We have changed the text as follows:

“Perceptual and affective components are conceptually different. The perceptual component refers to a disturbance in the perception of one’s own body; patients and healthy people exhibiting this component are unable to appraise their bodies accurately (11-13). For example, people with eating disorders and healthy individuals with more eating-disorder-related characteristics tend to overestimate their body size (i.e., there is a discrepancy between the perceived and actual self-body sizes) (11, 12, 14). The affective component relates to a disturbance in attitudes and feelings toward one’s own body. Patients and healthy people who exhibit the component are extremely dissatisfied with their body, i.e., their ideal and perceived body images differ considerably (11, 12, 14).” (pages 3-4, lines 59-68)

Comment 6

Similarly, the authors refer to eating-disorder tendencies instead of eating disorder characteristics or symptoms. The inconsistencies in terminology used in this manuscript versus those in published literature combined with the poor clarity of expression makes reading this section and understanding the goals of the study very confusing and difficult.

Reply

Thank you for your comment. We apologize for the use of confusing terminology and any lack of clarity. We have changed “eating disorder tendencies” to “eating-disorder characteristics”. We intended to use the phrase “people with greater eating disorder tendencies” to describe healthy people with eating-disorder symptoms and related personality traits. However, as the reviewer suggested, this term is confusing, so we have changed it. In the current study, we investigated both symptomatic traits (such as bulimia) and personality traits (such as ineffectiveness). Therefore, we have used the term “eating-disorder characteristics”, and added an explanation thereof to the text, as follows:

“Healthy individuals with greater body-image disturbance show higher levels of eating-disorder-related characteristics, such as interoceptive awareness, ineffectiveness, bulimia, body dissatisfaction, and drive for thinness (3, 4).” (page 3, lines 46-48)

Comment 7

The rationale for the study needs to be presented more strongly and clearly. The authors are basically replicating a very dated study (i.e., Probst et al., 1998) but using better methodology, measure of disordered eating and controlling for a wider range of possible confounding variables. But why is there a need to do this? 

Reply

Thank you for your comment. We apologize for the poor explanation of the rationale for this study. We have revised our manuscript to emphasize the study rationale. This study is novel because we evaluated the relationships between two distinct components of body-image disturbance and eating-disorder characteristics, which has not been done in previous studies. Our results help to explain the psychological mechanisms underlying eating-disorder characteristics and symptoms, and could aid the development of effective preventive and treatment strategies for eating disorders. We have added the following sentences:

“Understanding differences in the associations between eating-disorder characteristics and the two body-image-disturbance components would provide insight into the psychological mechanisms underlying eating-disorder characteristics, the mechanisms underlying eating-disorder symptoms, and preventive and treatment strategies for eating disorders.” (page 4, lines 79-83)

Comment 8

Has there been contention/debate in the literature about how body distortion versus body dissatisfaction are associated with different disordered eating characteristics/symptoms?

Reply

Thank you for your question. Some previous studies assumed that the affective component is related to concerns about body image. However, to the best of our knowledge, associations between the perceptual component and eating-disorder characteristics have not been addressed previously. We added the following text: 

“Despite the importance of body-image disturbance, little is known about how the two body-image disturbance components are related to various eating-disorder characteristics. No previous study has investigated the associations between the two components and eating-disorder characteristics; as the two relate conceptually to different mental activities, they may be differentially associated with different eating-disorder characteristics. The affective component is expected to be associated with body-image concerns (10, 16).” (page 4, lines 73-79)

Comment 9

Methodology: Rationale for recruiting University students needs to be provided.

Reply

Thank you for your suggestion. Several studies reported eating-disorder characteristics and body-image disturbance among university students. University students were recruited to allow comparison with previous studies. We have added the following sentences:

“It has been reported that some university students have eating-disorder characteristics and body-image disturbance (11, 12, 16). Many similar previous studies recruited healthy young adult women, including university students. Therefore, the study participants of this study were comparable with those of previous studies” (page 9, lines 185-189)

Comment 10

Moreover, why was the focus of judgment in the "Experiment" on the width of the body and not on the size of the body (i.e., “Is the width of the silhouette larger than your ideal image? versus "Is the silhouette larger than your ideal image?).

Reply

Thank you for your comment. We informed the participants that the width of the silhouettes was distorted. We showed participants black silhouette images, which differed only in width. We asked participants whether the width of the silhouette was larger than their actual or ideal body size. If we had asked participants to judge whether the size of the silhouette was larger than their actual or ideal image, the results would not have differed from when asking them regarding the width, because participants already knew that the silhouettes were changed only in terms of width. We now discuss this as follows: 

“The participants were informed in advance that they would be shown silhouette images of their body distorted only in terms of width, and that they would be required to estimate their actual and ideal body size according to the width of the images.” (page 12, lines 241-245)

Comment 11

Results: The Cronbach's alpha of the EDI-2 factors/subscales for the current sample needs to be reported either in the Method or Result section. Otherwise, the data appears to be appropriately analyzed.

Reply

Thank you for your comment and suggestion. As recommended by the reviewer, we have added the Cronbach’s alpha for EDI2 factors to the text and Table 1. (page 10)

We apologize for miscalculating one factor 6 item in the EDI2. After calculating the correlation coefficients, we neglected to reverse-score one item of factor 6. Therefore, we performed the analysis again and added the new results to Tables 1 and 3. 

Comment 12

However, the clarity of expression in this section can be improved. For example, why do we need the scatter plots? How do they assist in reporting the results?

Reply

Thank you for your question. We have added scatterplots showing the relationships between perceived–actual and perceived–ideal discrepancies (Fig 3), and between EDI2 factors and perceived–actual and perceived–ideal discrepancies (Fig 4). These scatterplots are informative regarding the distribution of the behavioral data, and demonstrate that the results were not due to outliers. We have added the following text to justify the use of scatterplots:

“The across-participants correlation between perceived–actual discrepancy and perceived–ideal discrepancy was not significant (r[27] = 0.10, p = 0.60); a lack of correlation between perceived–actual discrepancy and perceived–ideal discrepancy was observed in the scatterplot (Fig. 3).” (page 18, lines 383-386)

“These results are unlikely to be due to outliers (Fig. 4).” (page 20, line 413)

 In addition, we have improved the interpretation of our results via the following sentences: 

“These results suggest that, on average, participants perceived their body as 4.14% larger than their actual body, and their ideal body size was 5.78% smaller than their perceived body size.” (page 18, lines 381-383)

Comment 13

Discussion: While the findings of this study do confirm the findings of previous studies on the relationship between body image disturbances and disordered eating characteristics/symptoms, it is not clear what new insights this study provides. This needs to be clearly explained.

Reply

Thank you for your comment. The main novel finding of this study was that different eating-disorder characteristics were differentially associated with components of body-image disturbance in a healthy population. Our aim was not to replicate previous studies, as none of them reported eating-disorder characteristics specifically related to each component of body-image disturbance. To clarify the results of previous studies and gaps in knowledge, we have clarified our research question and novel findings as follows:

“The purpose of this study was to determine whether each of the two body-image-disturbance components were associated with distinct eating-disorder characteristics in healthy individuals.” (page 22, lines 436-438)

“This study is the first to report that the two components of body-image disturbance were associated with different EDI2 factors: perceived–actual discrepancy was correlated with EDI2 factor 3, which is related to negative self-evaluation, whereas perceived–ideal discrepancy was correlated with EDI2 factor 2, which is related to dissatisfaction with one’s own body. These results support the notion that the two components of body-image disturbance are independent (6, 14)” (page 23, lines 444-450)

Comment 14

Suggestion for future research are also needed.

Reply

Thank you for your comment. We have added suggestions for future research to the paragraph on limitations as follows:

“Although similarities between healthy people and patients have been reported (38, 44-46), patient studies are necessary to validate the relationships found in this study” (page 27, lines 541-543)

“Thus, our findings may be culture-specific, and additional international comparative research is needed to investigate this cultural specificity issue.” (page 27, lines 550-552)

Comment 15

I can sympathize and understand the difficulties in writing a manuscript in a language different to one's first language. I recommend that the authors get their manuscript checked by readers who are both proficient in English and familiar with the relevant literature.

Reply

Thank you for your comment. We have had the manuscript edited by an expert on eating disorders and an English editing service. 

 

Response to Reviewer 2

General

This manuscript was interesting to read, highlighting the need for researchers and clinicians to consider different aspects of body image. It uses a novel approach of measuring perceived body, and how aspects of body image are differentially related to factors in EDI2. As such, the manuscript have some important findings that should be of interest in the field of both eating disorders and body image. However, I believe that it needs some revisions and clarifications before it is ready for publication.

Reply

Thank you very much for reviewing our manuscript and providing insightful suggestions and comments. The reviewer raised concerns regarding the experimental methods, terminology, and manuscript structure. To address these, we have added details of the methods used to measure body-image disturbance, along with the reasons for choosing these methods. Second, we have revised some terms (such as “motivational component”) so that they accord with those used by previous studies. Finally, we have revised the structure of the manuscript in accordance with the reviewer’s suggestions.

Comment 1

One main concern is the mode of measurement. In the field of body image, there is a plethora of different measurements, stemming from the multidimensional construct of body image. Different researchers focus on different aspects of body image, thus measuring different constructs. The present study seems to be using an experimental mode of measurement. The measurement seems innovative, and do need to be more thoroughly introduced to the reader in the introduction. Why did authors choose this approach? How does it relate to other similar measurements? Is this a validation study?

Reply

Thank you for your suggestion and questions. As the reviewer pointed out, previous studies have described body-image disturbance in different ways. 

We chose the psychophysiological approach because it eliminates biases. Assessments of body image tend to be affected by social desirability bias; participants may overstate their perceived body size because they do not want to appear overconfident. We have added the following explanation for the use of the psychophysiological approach to the text: 

“We used a psychophysiological technique to reduce bias, such as social desirability bias (32), which affects responses on self-administered questionnaires.” (page 8, lines 157-159)

We chose the method of constant stimuli because this approach is convenient and associated with less bias compared to other psychophysiological techniques. A previous study (Probst et al., 1998) adopted a method of adjustment (i.e., participants adjusted images of their perceived and ideal bodies), in which participants could adjust their answers as they liked. Moreover, that design would incur an anchoring effect; participants’ perceived and ideal images could be influenced by the size of the body-image stimuli that they initially looked at. The use of the method of constant stimuli can avoid such biases. The stimuli were presented in a random order, which made it difficult for participants to premeditate their answers, and the random order of presentation avoided the anchoring effect. We justify the use of the method of constant stimuli as follows:

“The study participants adjusted images of their body themselves to match the ideal image on a life-sized screen. Therefore, participants could easily report that a silhouette image of a body corresponded to their own perceived body image, even if that was not the case. An anchoring effect may have been present (22). Individuals who initially view an image of an extremely overweight body will tend to perceive their own body as being larger (23).” (page 6, lines 122-128)

“To investigate this, we designed a forced-choice experiment based on the method of constant stimuli, in which participants viewed distorted silhouette images of their own photograph presented in a random order. We used a psychophysiological technique to reduce bias, such as social desirability bias (32), which affects responses on self-administered questionnaires. The participants were asked whether the body in the image was more overweight than their own or ideal body. Because the images were presented in a random order, the participants could not premeditate their answer (22). Moreover, the method of constant stimuli prevented the anchoring effect compared with other psychophysiological experiments.” (pages 7-8, lines 155-163)

This was not a validation study because no previous studies have investigated the relationships between eating-disorder characteristics and dissociable components of body-image disturbance using the same methods. We have added the following information to the text:

“Second, the study did not investigate the perceived–ideal discrepancy as a measure of the affective component (18). Rather, it investigated the relationships between the “actual–ideal” discrepancy and eating-disorder characteristics. However, the perceived–ideal discrepancy is now typically used as a measure of the affective component (12, 14).” (page 6, lines 128-132)

Comment 2

Another main concern regards the introduction, which I found a bit confusing. For instance, it is not clearly stated which of the cited references considers components of body image in patients with an eating disorder, and which are derived from a normal population. I found the same confusion in the term “eating-disorder tendencies”, is this part of eating disorder symptoms or disordered eating in the normal population? There is agreement between researchers that aspects of body image differ in individuals with or without an eating disorder. Thus, the introduction would benefit from separating research, or at least comment on the matter.

Reply

Thank you for your comment. As the reviewer suggested, we have clarified whether previous studies recruited patients or healthy people. The edited text now reads as follows:

“The perceptual component refers to a disturbance in the perception of one’s own body; patients and healthy people exhibiting this component are unable to appraise their bodies accurately (11-13).” (page 3, lines 59-62)

“Functional magnetic resonance imaging (fMRI) studies of people with eating disorders have indicated that two components of body-image disturbance relate to different phases of body-image processing: the perceptual component relates to visual processing (input), while the affective component relates to emotional processing (output) (6, 14, 15).” (page4, lines 68-72)

 We have changed “eating disorder tendencies” to “eating-disorder characteristics”. We used “eating disorder tendencies” to describe healthy people with eating-disorder symptoms and related personality traits. However, this terminology was deemed confusing, so we have changed it. We have added the following explanation of eating-disorder characteristics:

“Healthy individuals with greater body-image disturbance show higher levels of eating-disorder-related characteristics, such as interoceptive awareness, ineffectiveness, bulimia, body dissatisfaction, and drive for thinness (3, 4).” (page 3, lines 46-48)

Comment 3

Further, I have trouble with the choice of the term “motivation” as describing one of the body image constructs, which I am not familiar with. Could you provide a reference to this terminology? Or is the “affective” component you are referring to?

Reply

Thank you for your comment. We used the term “motivational component” to refer to the affective component. We agree that this term is confusing. Therefore, we have changed it to “affective component”, to match the terminology used in previous studies.

Comment 4

A third main concern regards the structure of the manuscript. In general, the manuscript is a bit confusing to read, where authors needs to rearrange some of the text. They are not following their own headings, mixing details between introduction and method. I will mention a few on the points below.

Reply

Thank you for your suggestion. We apologize for the unclear manuscript structure. In accordance with your recommendations, we have revised the structure.

Comment 5

The article begins with describing that body image disturbance consists of two components and cites three studies supporting this claim. When reading the studies, they do include more components- perceptive, affective, cognitive and a behavioral one. Body image is a complex structure, and your manuscript could be more clear about why you have selected these two components, or at least suggest these two are not the only components suggested in the literature.

Reply

Thank you for your suggestion. As the reviewer points out, “body image” is a multidimensional concept that includes a behavioral component. However, “disturbance in body-image processing (i.e., body-image disturbance)” comprises two or three components: perceptive (perceptual), affective, and cognitive. We focused on two of these components (perceptual and affective) in the current study because many previous studies considered the cognitive and affective components together (i.e., “cognitive-affective component”). Moreover, a review of fMRI studies of brain activation reported involvement of similar brain regions in the affective and cognitive components. Therefore, we focused on the perceptual and affective components only. We have added the following explanation to the manuscript:

“Accumulating evidence implies that body-image disturbance consists of perceptual and affective components (5-7). Several studies have also suggested that body-image disturbance has a cognitive component, which is sometimes considered to overlap with the affective component and is referred to as the “cognitive-affective component” (8-10). Moreover, a neuroimaging study of body-image disturbance suggested that the cognitive and affective components are not independent. That study categorized previous neuroimaging studies based on the body-image disturbance component that was evaluated, and reported that studies investigating the affective or cognitive component reported brain activation in similar regions (6).” (page 3, lines 49-58)

Comment 6

On page 5, critique is raised against a specific study. I think this could be stated in a more humble way, pointing to limitations raised in general. For instance, the notion on using an outdated version of EDI, for a study published in -98, while the present article uses a version of EDI that would also be considered obsolete in many instances of contemporary research.

Reply

Thank you for your comment. In accordance with the reviewer’s suggestion, we have modified our sentences to make our critique of the previous study more humble.

“The seminal study referred to above (18) was conducted around a quarter of a century ago; given recent advances in research techniques, the results seem to require updating in four respects: experimental design, conceptualization and measurement of disturbance, the version of the eating-disorder characteristics questionnaire used, and various confounding factors. First, the experimental design may have been susceptible to bias. The study participants adjusted images of their body themselves to match the ideal image on a life-sized screen. Therefore, participants could easily report that a silhouette image of a body corresponded to their own perceived body image, even if that was not the case. An anchoring effect may have been present (22). Individuals who initially view an image of an extremely overweight body will tend to perceive their own body as being larger (23). Second, the study did not investigate the perceived–ideal discrepancy as a measure of the affective component (18). Rather, it investigated the relationships between the “actual–ideal” discrepancy and eating-disorder characteristics. However, the perceived–ideal discrepancy is now typically used as a measure of the affective component (12, 14). The previous study reported a significant correlation between “actual–ideal” discrepancy and body dissatisfaction (EDI subscale score) in healthy controls, but not patients. However, there may be a significant correlation between “perceived–ideal” discrepancy and body dissatisfaction in eating-disorder patients. Third, as an older work, the previous study used the original EDI; newer versions are now available, i.e., the EDI2 and 3 (18, 21, 24), which include an additional 27 items (three subscales). Furthermore, the previous study did not observe any correlations between perceived–actual discrepancy and eating-disorder characteristics in healthy people. It is possible that a correlation would be detected using the newer versions of the EDI. Finally, potential confounders, such as body mass index (BMI), were not considered. BMI is associated with both perceived–actual and perceived–ideal discrepancies (12, 25, 26). Healthy people with a higher BMI are more prone to body dissatisfaction (27-29). Therefore, BMI could have influenced the previous study’s results, in addition to the menstrual cycle and body checking. Several previous studies suggested changes in body size estimation and satisfaction in female patients before and during menstruation (30, 31). Some people use mirrors to check body size daily, whereas others do not; this could influence the accuracy of body size estimations.” (pages 6-7, lines 117-149)

 We have also added an explanation regarding our use of the EDI2 to the manuscript (see comment 9). The EDI2 is the most recent version of the EDI with a Japanese translation. We have added the following explanation to the article: 

“Third, we used the latest version of the Japanese EDI2 to evaluate eating-disorder characteristics (the EDI3 was not available in Japanese at the time of the experiments) (20, 21).” (page 8, lines165-167)

Comment 7

Aim. The aim would benefit from a definition of the term “eating-disorder tendencies” in the introduction.

Reply

Thank you for your suggestion. In accordance with comment 2, “eating-disorder tendencies” has been changed to “eating-disorder characteristics.” (c.f. Comment 2)

Comment 8

The sentence following the aim seems to describe methodology, would you consider moving it for clarity? Thus, the stated hypothesizes (“we predicted”-change to hypothezised?) directly follows on the aim. Moreover, the description on the Japanese version of EDI2 would benefit of a move to earlier in the introduction, with a description on the factors identified and what you believe is important for your study.

Reply

Thank you for your suggestion. As suggested, we have added the hypothesis to the paragraph after the aims (page 7, lines 152-154). We have also moved the description of the structure of the Japanese version of the EDI2 to an earlier paragraph in the Introduction (page 5, lines 98-106).

Comment 9

Further, since you have described that an EDI3 exists, why did you settle for EDI2?

Reply

We apologize for the lack of clarity regarding the EDI version used (see comment 6). We have added the following text:

“Third, we used the latest version of the Japanese EDI2 to evaluate eating-disorder characteristics (the EDI3 was not available in Japanese at the time of the experiments) (20, 21).” (page 8, lines165-167)

Comment 10

Page 7, the last two sentences in the introduction. These are very unclear, I do not understand what you are trying to say, please read these and consider rephrasing.

Reply

We apologize for the unclear text. We intended to state our more concrete hypothesis, that EDI2 factors would correlate with each discrepancy. To clarify that these sentences comprise our hypothesis, we have changed them as follows.: 

“We hypothesized that perceived–actual discrepancy correlates with factors 3, 4, and 6 because these factors include items originally classified as ineffectiveness (20, 21). Meanwhile, perceived–ideal discrepancy correlates with factors 1 and 2 because these factors include items originally classified as body dissatisfaction and drive for thinness (20, 21).” (page 8, lines 170-174)

Comment 11

Method. Under the heading of participants, you could consider the number and characteristics of the sample as part of your results. I could not find any descriptions on how participants were recruited, were they recommended participation, or did they respond to advertisement?

Reply

Participants were recruited via an advertisement displayed at the university. We have added the following explanation to the article:

“Participants were invited to participate in this study via an advertisement displayed in the university office.” (page 9, lines 189-190)

Comment 12

Measures. In the description on quantification (page 13), adding possible ratings that participants could receive would aid in understanding your scale. Without this clarification, your results in Table 2 are hard to understand and interpret. For instance, how should I interpret the mean of 4.14? That the mean percentage of discrepancy is 4.14?

Reply

In accordance with your suggestion, we have added the following example ratings: 

“For example, if a participant’s perceived–actual discrepancy was 5, the participant overestimated her body size by 5%.” (page 15, lines 306-308)

“Similarly, if a participant’s perceived–ideal discrepancy was 5, the participant wishes to be 5% thinner than her perceived self-image” (page 15, lines 310-311)

 Moreover, as it seems that our results were somewhat difficult to understand, we have added the following interpretation: 

“These results suggest that, on average, participants perceived their body as 4.14% larger than their actual body, and their ideal body size was 5.78% smaller than their perceived body size.” (page 18, lines 381-383)

Comment 13

2.4.2 Questionnaire. EDI2 is described as distinguishing patients from controls-is it patients with an eating disorder, or another patient-group?

Reply

We apologize for the insufficient explanation. We intended to convey that the EDI2 scores of the patients were significantly higher compared to those of the healthy people. Additionally, the EDI2 scores of the healthy people with eating-disorder symptoms were significantly higher compared to those of healthy people without symptoms. However, the EDI2 cannot distinguish patients from healthy people, as there is no validated cut-off value. We have revised the text to clarify these points, as follows: 

“The participants were informed in advance that they would be shown silhouette images of their body distorted only in terms of width, and that they would be required to estimate their actual and ideal body size according to the width of the images.” (page 12, lines 241-245)

Comment 14

Is there a cut-off score separating patients from healthy controls? Stating this will make it possible to judge if your study population is indeed healthy, at risk, or should even be considered as showing ratings indicative of an eating disorder. (again, this points as to why table 1 should be part of your results)

Reply

Thank you for your suggestion. As explained in the response to comment 13, the EDI2 does not have a validated cut-off score for distinguishing between healthy and at-risk populations.

Comment 15

2.4.4. Menstrual cycle. In the section on BMI you describe how you measured it. I would suggest using this way of describing measurement in the method section, and moving descriptions and citations to either the introduction or discussion. Thus, first sentence in this heading would benefit from a removal.

Reply

In accordance with the reviewer’s suggestion, we have moved the explanation of the need to exclude the effects of menstrual cycle from the Methods to the Introduction. The edited sentence reads as follows:

“Several previous studies suggested changes in body size estimation and satisfaction in female patients before and during menstruation (30, 31)” (page 7, lines 146-148)

Comment 16

2.4.5. Please see comment above.

Reply

We have moved the explanation of the need to exclude the effects of daily body-checking behavior from the Methods to the Introduction. The edited sentence reads as follows:

“Some people use mirrors to check body size daily, whereas others do not; this could influence the accuracy of body size estimations.” (page 7, lines 148-149)

Comment 17

Results. 3.1. Mean levels reported needs to clarified, what are these numbers representing?

Reply

Thank you for your comment. We have clarified the interpretation of our results (see reply to Comment 12) as follows:

“These results suggest that, on average, participants perceived their body as 4.14% larger than their actual body, and their ideal body size was 5.78% smaller than their perceived body size.” (page 18, lines 381-383)

Comment 18

3.1. Here you state the correlation between perceived-actual and perceived-ideal (page 16), which is stated again on page 17?

Reply

We apologize for the unclear manuscript structure. The reviewer may have been referring to this sentence “The correlation between perceived–actual discrepancy and perceived–ideal discrepancy was not significant (df = 27, r = 0.10, p = 0.60).” (page 19, lines 396-397). This sentence was part of the legend for Fig. 3; we have moved this sentence to just after the Fig. 3 caption to make it clear.

Comment 19

Table 3. Why are not all correlations between the different body image components and factors reported? The supporting information S1 Table uses abbreviations that are uninterpretable.

Reply

We have added all of the correlations to Table 3. Previously, we only showed correlations of certain factors because our a priori hypothesis concerned the correlations of EDI2 factors with perceived–actual and perceived–ideal discrepancies. We have added the correlations and uncorrected p-values for all factors. Asterisks denote correlations that remained significant after correcting for multiple comparisons.

 We apologize for using unclear abbreviations in Table S1, which has been omitted from the revised manuscript.

Comment 20

Discussion. I found the discussion a bit confusing as well, what are the main findings in the current study?

Reply

We apologize for the unclear text. The main findings of the current study were that each component of body-image disturbance was associated with different eating-disorder characteristics. To clarify our main findings, we have rearticulated our research question and main findings at the beginning of the Discussion. The edited text reads as follows: 

“The purpose of this study was to determine whether each of the two body-image-disturbance components were associated with distinct eating-disorder characteristics in healthy individuals.” (page 22, lines 436-438)

“This study is the first to report that the two components of body-image disturbance were associated with different EDI2 factors: perceived–actual discrepancy was correlated with EDI2 factor 3, which is related to negative self-evaluation, whereas perceived–ideal discrepancy was correlated with EDI2 factor 2, which is related to dissatisfaction with one’s own body.” (page 23, lines 444-449)

Comment 21

Page 20. The paragraph beginning in line 4. Here authors introduce attentional bias as an explanation. This is a bit speculative, and not part of the study results. Again, this paragraph lacks clarity, and as a reader I am confused as to how your results “implies that attentional bias plays a mediating role…”?

Reply

Thank you for your comment. We believe that our results are consistent with those of previous studies (i.e., perceived–actual discrepancy is related to attentional bias). We have described the hypothesis about this relationship cited in previous studies as follows:

“The observed positive correlation between perceived–actual discrepancy and negative self-evaluation (factor 3) supports a previous hypothesis that attentional bias underlies perceived–actual discrepancy. We showed that, among the three EDI2 domains of ineffectiveness, perceived–actual discrepancy was related to negative self-evaluation rather than maturity fear and confusion of the mind (factor 4) or desire for achievement (factor 6). Previous studies reported that attentional bias toward the waist was associated with perceived–actual discrepancy in a healthy population (13, 17). In particular, healthy people with greater perceived–actual discrepancy exhibited attentional bias toward one side of the torso (17). Previous studies also reported a relationship between negative self-evaluation and attentional bias. Low self-esteem has been proposed as one of the reasons for attentional bias toward disliked parts of one’s own body (40). We observed a relationship between perceived–actual discrepancy and negative self-evaluation, which bridges the results of previous studies and supports the hypothesis that attentional bias underlies perceived–actual discrepancy.” (page 23, lines 451-465)

 As pointed out by the reviewer, our results do not demonstrate a mediating role of attentional bias. Therefore, we could not draw conclusions regarding the role of attentional bias. The text has been edited accordingly, as follows: 

“This study implies an involvement of attentional bias; future studies should further explore this possibility.” (page 24, lines 470-472)

Comment 22

Page 21, first sentence. Again, it is unclear, how does this “reflect an eating-disorder-related phenomenon”? You have not investigated patients with an eating disorder, nor reported any pathological tendencies in the study population.

Reply

Thank you for your comment. We intended to convey that the positive correlation between perceived–ideal discrepancy and dissatisfaction toward one’s body may reflect negative feelings toward the body (similar to that observed in people with eating disorders). This positive correlation suggests that participants with greater perceived–ideal discrepancy tended to be dissatisfied with their own bodies, regardless of actual body shape and weight. We believe that this dissatisfaction is independent of the actual body, and is not a result of a healthy mental outlook. Therefore, we used the following phrase: “reflect an eating-disorder-related phenomenon”. However, we agree that this expression is confusing, so have edited it as follows: 

“Although overweight people tend to be dissatisfied with their bodies (27-29), the correlation remained significant after controlling for BMI. Irrespective of their actual body shape and weight, participants with a greater perceived–ideal discrepancy were less satisfied with their body. This suggests that the perceived–ideal discrepancy reflects body dissatisfaction independent of actual body weight and shape.” (page 24, lines 477-482)

Comment 23

In the middle of page 21, one sentence starts with “This unreasonable body dissatisfaction”. Do you consider how a healthy individual perceives his/her body as target for judgment? Is there a wrong and right aspect to this? I think it is more humble to use your own terminology here, on discrepancy.

Reply

Thank you for your comment. As the reviewer points out, the term “unreasonable” is not appropriate; the terms “accurate” and “inaccurate” should be used. We used the term “unreasonable” to describe participants with greater perceived–ideal discrepancy who exhibited body dissatisfaction independent of the actual body size. We believe that the term “independent” is more appropriate in this context. We have deleted the term “unreasonable”, and also added a description of the incongruency between actual body weight/shape and the magnitude of body dissatisfaction, as follows:

“Irrespective of their actual body shape and weight, participants with a greater perceived–ideal discrepancy were less satisfied with their body. This suggests that the perceived–ideal discrepancy reflects body dissatisfaction independent of actual body weight and shape.” (page 24, lines 479-482)

 In addition, we have revised certain expressions used in the paper, and split the paragraphs to clarify the meaning of the term “independent”. In the first paragraph, we have discussed the interpretation of our findings (correlation between perceived–ideal discrepancy and EDI2 factor 2). In the second paragraph, we have added a discussion of the correlation between perceived–ideal discrepancy and factor 2.

“Based on the above, body dissatisfaction independent of actual body weight and shape may result from a deficit in a feedback circuit between change in body dissatisfaction and change in actual body weight and size. Most, but not all, overweight people report less dissatisfaction with their body after weight loss. Body dissatisfaction independent of the actual body weight and size may explain why some people persist with a diet. This is consistent with a previous study that reported that some healthy people with higher body dissatisfaction and a low BMI continue to diet (3). In contrast to the relationship between perceived–ideal discrepancy and factor 2, the correlation between perceived–ideal discrepancy and factor 1 was not significant after controlling for BMI, which suggests that this correlation was strongly influenced by participants’ actual body weight and shape. Another possible reason for the lack of a significant correlation between perceived–ideal discrepancy and factor 1 is that factor 1 also includes items related to bulimia (except for those related to body weight and shape). A longitudinal study is needed to confirm whether the perceived–ideal discrepancy reflects refractory body dissatisfaction independent of actual body weight and shape change.” (pages 24-25, lines 483-499)

Comment 24

Page 21, last sentences. Here you present results that have not been reported, and again, as a reader I am confused as to why only BMI is selected and not all factors you are controlling for. I don’t understand how this implies a mediation by BMI.

Reply

Thank you for your comment. The role of mediation is not clear. Therefore, we modified the sentence to state that BMI may have influenced the correlation between perceived–ideal discrepancy and factor 1. 

We apologize for not explaining why we focused on BMI only. Overweight people tend to have higher levels of body dissatisfaction, which is why we focused on BMI. We have added an explanation of this to the article, and have also discussed the modulatory effects of BMI on the relationship between perceived–ideal discrepancy and factor 1. We have edited the text as follows: 

“In contrast to the relationship between perceived–ideal discrepancy and factor 2, the correlation between perceived–ideal discrepancy and factor 1 was not significant after controlling for BMI, which suggests that this correlation was strongly influenced by participants’ actual body weight and shape. Another possible reason for the lack of a significant correlation between perceived–ideal discrepancy and factor 1 is that factor 1 also includes items related to bulimia (except for those related to body weight and shape). A longitudinal study is needed to confirm whether the perceived–ideal discrepancy reflects refractory body dissatisfaction independent of actual body weight and shape change.” (page 25, lines 490-499)

Moreover, we deleted Table S2 because it was deemed confusing.

Comment 25

page 23, the paragraph on the middle of the page. This sentence was confusing to read, this hypothesis was not part of the current study? Consider rephrasing.

Reply

We apologize for the unclear explanation. Our results support a relationship between perceived–actual discrepancy and attentional bias, similar to that reported in previous studies. The aim of this paragraph was to discuss the hypothesis generated in previous studies. We also emphasized that our results support the previous hypothesis. However, some sentences in this paragraph were unclear, so have been revised. The edited sentences read as follows:

“Our results supported a relationship between perceived–actual discrepancy and attentional bias, which may be clinically relevant since it is in line with interventions suggested in recent studies. One study reported that a full-body illusion created using virtual-reality techniques improved perceived–actual discrepancy (42). Another study reported that a virtual reality-based body exposure therapy reduced attentional bias (43). Decreasing attentional bias may improve perceived–actual discrepancy, and virtual-reality techniques may be one of the most effective ways to decrease attentional bias toward particular body parts. An interventional study is needed to investigate the relationship between improved perceived–actual discrepancy and attentional bias.” (pages 26-27, lines 529-538)

Comment 26

Conclusions. Again, is attentional bias part of the current study? Otherwise I don’t think it should be part of your conclusions.

Reply

Thank you for your comment. As pointed out by the reviewer, attentional bias was not assessed in the current study. Therefore, we have eliminated the sentence on attentional bias from the conclusions.

---

## [Decision Letter · Decision Letter 1]

18 Oct 2021

PONE-D-21-08191R1Two components of body-image disturbance are differentially associated with distinct eating disorder characteristics in healthy young womenPLOS ONE

Dear Dr. Hamamoto,

Thank you for submitting your manuscript to PLOS ONE. After careful consideration, we feel that it has merit but does not fully meet PLOS ONE’s publication criteria as it currently stands. Therefore, we invite you to submit a revised version of the manuscript that addresses the points raised during the review process.

Specifically, please address the minor revisions from Reviewer #2.

We look forward to receiving your revised manuscript.

Kind regards,

Cherilyn N. McLester, PhD

Academic Editor

PLOS ONE

Journal Requirements:

Reviewers' comments:

Reviewer's Responses to Questions

**Comments to the Author**

1. If the authors have adequately addressed your comments raised in a previous round of review and you feel that this manuscript is now acceptable for publication, you may indicate that here to bypass the “Comments to the Author” section, enter your conflict of interest statement in the “Confidential to Editor” section, and submit your "Accept" recommendation.

Reviewer #1: All comments have been addressed

Reviewer #2: All comments have been addressed

2. Is the manuscript technically sound, and do the data support the conclusions?

Reviewer #1: Yes

Reviewer #2: Yes

3. Has the statistical analysis been performed appropriately and rigorously? 

Reviewer #1: Yes

Reviewer #2: Yes

4. Have the authors made all data underlying the findings in their manuscript fully available?

Reviewer #1: Yes

Reviewer #2: Yes

5. Is the manuscript presented in an intelligible fashion and written in standard English?

Reviewer #1: Yes

Reviewer #2: Yes

6. Review Comments to the Author

Reviewer #1: Thank you for carefully considering my feedback comments. All my comments have been sufficiently addressed by the authors. All the best with your research.

Reviewer #2: It is clear that authors have taken time and made an effort to improve the manuscript, and have done a great job in giving attention to all comments. I reread the entire manuscript first, and the improvements are impressive! The flow of the language is better, and descriptions have been more clearly stated, making the manuscript and the investigation an interesting article to read.

When looking at the specific comments from the former review, there are just some minor remarks that you can consider to revise:

• I think your aim could be even more to the point if you include EDI-2 in your first sentence of the aim, p.7 line 149-151.

• Your reply to comment 22, p.24, line 477-482, could these results be linked to low variation in BMI in your study group? Where there any overweight or obese individuals in your study group?

Two comments that were not included in first review, and became apparent upon second review:

• In the method section, I could not find information on when EDI-2 was administered, before or after the experiment?

• One limitation worth mentioning is the limited sample size, could this have any implications for your results or generalizability?

Again, please see these comments as suggestions for improvements.

7. PLOS authors have the option to publish the peer review history of their article (what does this mean?). If published, this will include your full peer review and any attached files.

Reviewer #1: No

Reviewer #2: **Yes: **Maria Fogelkvist

---

## [Author Response · Author response to Decision Letter 1]

22 Nov 2021

Dear Editor, PLOS ONE,

Thank you for giving us the opportunity to submit our revised manuscript entitled “Two components of body-image disturbance are differentially associated with distinct eating disorder characteristics in healthy young women” to PLOS ONE. We appreciate your comments and those of the reviewers, and have revised the manuscript accordingly.

Our responses to the comments by the reviewers are as follows:

Response to Reviewer 1

General

Thank you for carefully considering my feedback comments. All my comments have been sufficiently addressed by the authors. All the best with your research.

Reply

Thank you for carefully reviewing our manuscript again. We appreciate your insightful comments.

Response to Reviewer 2

General

It is clear that authors have taken time and made an effort to improve the manuscript, and have done a great job in giving attention to all comments. I reread the entire manuscript first, and the improvements are impressive! The flow of the language is better, and descriptions have been more clearly stated, making the manuscript and the investigation an interesting article to read.

Reply

Thank you for carefully reviewing our manuscript again and providing insightful suggestions and comments. We revised the manuscript according to the reviewer’s suggestions to enhance readability.

Comment 1

When looking at the specific comments from the former review, there are just some minor remarks that you can consider to revise:

• I think your aim could be even more to the point if you include EDI-2 in your first sentence of the aim, p.7 line 149-151.

Reply

Thank you for your suggestion. As the reviewer suggested, we included EDI-2 in the first sentence of the study aims part. The edited text reads as follows:

“The present study aimed to determine the relationships between two body-image-disturbance components (perceptual and affective components) and eating-disorder characteristics represented by EDI2 scores.” (page 7, lines 149-151)

Comment 2

• Your reply to comment 22, p.24, line 477-482, could these results be linked to low variation in BMI in your study group? Where there any overweight or obese individuals in your study group?

Reply

The BMI in our study ranged from 17.58 to 24.87 (i.e., there were no obese individuals [BMI > 25.0 kg/m2]). Although this BMI range is consistent with that of Japanese females in their twenties, it is slightly lower than that in other countries.

We believe that the variation in BMI in our study was sufficiently informative given its significant correlation with dissatisfaction toward one’s body (factor 2 of EDI2) and perceived–ideal discrepancy. In the revised manuscript, we have discussed the role of BMI in the relationship between perceived–ideal discrepancy and dissatisfaction toward one’s body, as follows:

“We also observed that BMI was strongly correlated with dissatisfaction toward one’s body (r[27] = 0.71, t = 5.22, p < 0.001) and perceived–ideal discrepancy (r[26] = 0.61, t =3.94, p < 0.001). However, the correlation between perceived–ideal discrepancy and dissatisfaction toward one’s body remained significant after controlling for BMI.” (page 24, lines 478-482)

The fact that all of the participants were underweight or had a normal BMI (17.58–24.87) has been cited as one of the limitations of the study, i.e., the questionable generalizability of our results, as pointed out by the reviewer in Comment 4. The lack of obese participants was not surprising because the prevalence of obesity in Japanese females in their twenties is only 8.9%. The mean BMI in the current study was similar to that in previous studies conducted in Japan, but lower than that in previous studies conducted in other countries. We have added the following sentences to the limitations:

“Second, our results have limited cross-national generalizability. The BMI of the participants in the current study was similar to that in previous body-image studies conducted in Japan (47, 48), but lower than that in studies conducted in Western countries (12, 13). The lack of obese individuals (BMI > 25.0 kg/m2) in the current study is explained by the low prevalence of obesity in Japanese females in their twenties (i.e., 8.9% according to a national survey conducted in 2019 by the Ministry of Health, Labor and Welfare). BMI is a critical factor in body-image studies because it is strongly correlated with concerns about body image (12, 27-29). The comparability of our findings with those of studies that recruited participants from different cultures is also limited by our use of the Japanese version of the EDI2, which has a culture-specific six-factor structure. However, the results for factor 2 may be directly comparable to those of other studies because this factor is primarily composed of body-dissatisfaction items from the original EDI2 subscale. In contrast, factor 3 is composed of items from four subscales of the original EDI2: ineffectiveness, interoceptive awareness, impulse regulation, and social insecurity. In addition, this study had a small sample, although it was based on a power analysis. Due to the small sample size, relationships between the two components of body-image disturbance and eating disorder characteristics with small effect sizes may have been missed. Biological and cultural factors reflected in the lower BMI range and structure of the EDI2 would have influenced our findings and limit the generalizability of the current study. International comparative studies including larger samples and wider ranges of BMI will be needed to confirm our results.” (pages 27-28, lines 547-569)

Comment 3

Two comments that were not included in first review, and became apparent upon second review:

• In the method section, I could not find information on when EDI-2 was administered, before or after the experiment?

Reply

We apologize for the poor description of the procedure. We administered the EDI-2 before the experiment to avoid any effects of the tasks on the questionnaire results. We have added the following description:

“After completing the EDI2 questionnaire, participants performed the actual-body and ideal-body tasks (Fig. 1).” (page 12, lines 240-241)

“The participants completed the EDI2 (20) before performing the actual-body and ideal-body tasks.” (page 15, lines 314-315)

Comment 4

• One limitation worth mentioning is the limited sample size, could this have any implications for your results or generalizability?

Reply

Thank you for your suggestion. As pointed out by the reviewer, the sample size was small; however, it was adequate to detect the relationships of interest according to the power analysis. However, we may not have detected relationships between the two components of body-image disturbance and eating-disorder characteristics with small effect sizes, which is one of the study limitations. The implications of the small sample size for generalizability are discussed in the limitations section. We have added the following sentences:

“Second, our results have limited cross-national generalizability. The BMI of the participants in the current study was similar to that in previous body-image studies conducted in Japan (47, 48), but lower than that in studies conducted in Western countries (12, 13). The lack of obese individuals (BMI > 25.0 kg/m2) in the current study is explained by the low prevalence of obesity in Japanese females in their twenties (i.e., 8.9% according to a national survey conducted in 2019 by the Ministry of Health, Labor and Welfare). BMI is a critical factor in body-image studies because it is strongly correlated with concerns about body image (12, 27-29). The comparability of our findings with those of studies that recruited participants from different cultures is also limited by our use of the Japanese version of the EDI2, which has a culture-specific six-factor structure. However, the results for factor 2 may be directly comparable to those of other studies because this factor is primarily composed of body-dissatisfaction items from the original EDI2 subscale. In contrast, factor 3 is composed of items from four subscales of the original EDI2: ineffectiveness, interoceptive awareness, impulse regulation, and social insecurity. In addition, this study had a small sample, although it was based on a power analysis. Due to the small sample size, relationships between the two components of body-image disturbance and eating disorder characteristics with small effect sizes may have been missed. Biological and cultural factors reflected in the lower BMI range and structure of the EDI2 would have influenced our findings and limit the generalizability of the current study. International comparative studies including larger samples and wider ranges of BMI will be needed to confirm our results.” (pages 27-28, lines 547-569)

---

## [Decision Letter · Decision Letter 2]

28 Dec 2021

Two components of body-image disturbance are differentially associated with distinct eating disorder characteristics in healthy young women

PONE-D-21-08191R2

Dear Dr. Hamamoto,

We’re pleased to inform you that your manuscript has been judged scientifically suitable for publication and will be formally accepted for publication once it meets all outstanding technical requirements.

Kind regards,

Cherilyn N. McLester, PhD

Academic Editor

PLOS ONE

Additional Editor Comments (optional):

Reviewers' comments:

Reviewer's Responses to Questions

**Comments to the Author**

1. If the authors have adequately addressed your comments raised in a previous round of review and you feel that this manuscript is now acceptable for publication, you may indicate that here to bypass the “Comments to the Author” section, enter your conflict of interest statement in the “Confidential to Editor” section, and submit your "Accept" recommendation.

Reviewer #2: All comments have been addressed

2. Is the manuscript technically sound, and do the data support the conclusions?

Reviewer #2: Yes

3. Has the statistical analysis been performed appropriately and rigorously? 

Reviewer #2: Yes

4. Have the authors made all data underlying the findings in their manuscript fully available?

Reviewer #2: Yes

5. Is the manuscript presented in an intelligible fashion and written in standard English?

Reviewer #2: Yes

6. Review Comments to the Author

Reviewer #2: Thank you for your patience and for giving attention to all my comments. I have no further comments, and recommend the article for accept.

7. PLOS authors have the option to publish the peer review history of their article (what does this mean?). If published, this will include your full peer review and any attached files.

Reviewer #2: **Yes: **Maria Fogelkvist

---

## [Editor Report · Acceptance letter]

3 Jan 2022

PONE-D-21-08191R2 

Two components of body-image disturbance are differentially associated with distinct eating disorder characteristics in healthy young women 

Dear Dr. Hamamoto:

I'm pleased to inform you that your manuscript has been deemed suitable for publication in PLOS ONE. Congratulations! Your manuscript is now with our production department. 

Kind regards, 

on behalf of

Dr. Cherilyn N. McLester 

Academic Editor

PLOS ONE